# Aberrant enhancer hypomethylation contributes to hepatic carcinogenesis through global transcriptional reprogramming

Lei Xiong[1,2], Feng Wu[1,2], Qiong Wu[2], Liangliang Xu[2], Otto K. Cheung[2], Wei Kang[1], Myth T. Mok[2], Lemuel L. M. Szeto[2], Cheuk-Yin Lun[2], Raymond W. Lung[1], Jinglin Zhang[1], Ken H. Yu[1,3], Sau-Dan Lee[3], Guangcun Huang[4], Chiou-Miin Wang[4], Joseph Liu[4], Zhuo Yu[5], Dae-Yeul Yu[6], Jian-Liang Chou[7], Wan-Hong Huang[7], Bo Feng[2], Yue-Sun Cheung[8], Paul B. Lai[8], Patrick Tan[9,10], Nathalie Wong[1], Michael W. Chan[7], Tim H. Huang[4], Kevin Y. Yip[3], Alfred S. Cheng[2] & Ka-Fai To[1,11]

Hepatocellular carcinomas (HCC) exhibit distinct promoter hypermethylation patterns, but the epigenetic regulation and function of transcriptional enhancers remain unclear. Here, our affinity- and bisulfite-based whole-genome sequencing analyses reveal global enhancer hypomethylation in human HCCs. Integrative epigenomic characterization further pinpoints a recurrent hypomethylated enhancer of CCAAT/enhancer-binding protein-beta (C/EBPβ) which correlates with *C/EBPβ* over-expression and poorer prognosis of patients. Demethylation of *C/EBPβ* enhancer reactivates a self-reinforcing enhancer-target loop via direct transcriptional up-regulation of enhancer RNA. Conversely, deletion of this enhancer via CRISPR/Cas9 reduces C/EBPβ expression and its genome-wide co-occupancy with BRD4 at H3K27ac-marked enhancers and super-enhancers, leading to drastic suppression of driver oncogenes and HCC tumorigenicity. Hepatitis B X protein transgenic mouse model of HCC recapitulates this paradigm, as *C/ebpβ* enhancer hypomethylation associates with oncogenic activation in early tumorigenesis. These results support a causal link between aberrant enhancer hypomethylation and *C/EBPβ* over-expression, thereby contributing to hepato-carcinogenesis through global transcriptional reprogramming.

[1] Department of Anatomical and Cellular Pathology, The Chinese University of Hong Kong, Hong Kong SAR, China. [2] School of Biomedical Sciences, The Chinese University of Hong Kong, Hong Kong SAR, China. [3] Department of Computer Science and Engineering, The Chinese University of Hong Kong, Hong Kong SAR, China. [4] Department of Molecular Medicine, The University of Texas Health Science Center at San Antonio, San Antonio, TX 78245, USA. [5] Department of Liver Disease, Shuguang Hospital affiliated to Shanghai University of Traditional Chinese Medicine, Shanghai 201203, China. [6] Disease Model Research Laboratory, Genome Editing Research Center, Korea Research Institute of Bioscience and Biotechnology, Daejeon 305-806, Republic of Korea. [7] Department of Biomedical Sciences, National Chung Cheng University, Chia-Yi 62102, Taiwan, Republic of China. [8] Department of Surgery, The Chinese University of Hong Kong, Hong Kong SAR, China. [9] Program in Cancer and Stem Cell Biology, Duke-NUS Medical School, Singapore 169857, Singapore. [10] Genome Institute of Singapore, Singapore 138672, Singapore. [11] State Key Laboratory of Translational Oncology, The Chinese University of Hong Kong, Hong Kong SAR, China. Correspondence and requests for materials should be addressed to K.Y.Y. (email: kevinyip@cse.cuhk.edu.hk) or to A.S.C. (email: alfredcheng@cuhk.edu.hk) or to K.-F.T. (email: kfto@cuhk.edu.hk)

Hepatocellular carcinoma (HCC) is the third leading cause of global cancer-related deaths with an annual incidence rate of approximately 850,000 cases[1,2]. Major risk factors for HCC include chronic hepatitis B virus (HBV) and hepatitis C virus (HCV) infections, alcohol abuse, and non-alcoholic fatty liver disease (NAFLD) associated with obesity and diabetes[1,2]. Genetic and epigenetic alterations that progressively accumulate in the chronic inflammatory milieu lead to the initiation and progression of HCC, but the precise molecular events are only partially understood[3]. Although much effort has been devoted to elucidating the genetic defects underlying malignant transformation of hepatocytes, only few druggable driver mutations have been revealed for a fraction of HCC patients due to tumor heterogeneity[2,3]. Given the marginal benefit of the small-molecule agent sorafenib and failure of multiple molecular drugs in phase III trials for HCC patients[2], HCC methylome profiling would uncover the epigenetic vulnerabilities for the development of new intervention strategies.

Epigenome disruption has emerged as a major hallmark in HCC as revealed by the discoveries of somatic mutations in chromatin regulators and numerous epigenetic abnormalities[4,5]. Chronic hepatitis infection and NAFLD have been shown to induce aberrant DNA methylation that may contribute to the development of HCC[6–8]. Genome-scale DNA methylation profiles of nearly two-hundred HCC cases via an array-based platform further revealed distinct cancer-specific DNA hypermethylation clusters[9]. Most of these studies, however, focused on altered methylation at gene promoters and CpG islands/shores. Apart from the classical promoter hypermethylation-mediated gene silencing, the epigenetic regulation and function of distal *cis*-regulatory regions have yet to be elucidated.

Transcriptional enhancers are distal non-coding regions crucial for cell identify specification. These key regulatory elements are driven by combinatorial assembly of lineage-determining transcription factors, coactivators and bromodomain and extra-terminal domain (BET) family proteins including BRD4, which recruits transcriptional complexes to acetylated lysine 27 of histone H3 (H3K27ac) for enhancer RNA (eRNA) synthesis[10,11]. Accumulating evidence has shown the importance of enhancers and super-enhancers, i.e., clusters of aberrantly active enhancers that are strongly enriched by BRD4 and H3K27ac, in dysregulated expression of oncogenes[12,13]. In addition, gene dysregulation is more correlated with altered methylation at their enhancers than promoters in many transformed cell types, including those of hepatocyte origin[14–16].

We have recently developed a new method for inferring enhancer-target interactions by integrating epigenomic and transcriptomic data from hundreds of primary cells and tissues, which enabled the identification of target genes that are specifically controlled by differentially methylated enhancers (DMEs) in HCC cells[17]. Utilizing affinity- and bisulfite-based whole-genome sequencing, here we show a genome-wide enhancer hypomethylation pattern in primary human HCCs. Our integrative epigenomic analysis highlights a recurrent hypomethylated enhancer of CCAAT/enhancer-binding protein-beta (*C/EBPβ*), which exhibits clinical and biological significance in promoting HCC tumorigenicity through global transcriptional reprogramming.

## Results

**Integrative epigenomic analysis of human HCCs.** We performed methyl-binding DNA capture sequencing (MBDCap-seq)[18] on 33 pairs of HCC tumors and matched non-tumor tissues from the same patients. From each sample, we obtained on average 51,855,400 sequencing tags that were aligned to the human reference genome hg19. Globally, a clear hypomethylation

pattern in the tumor group was observed when compared with the non-tumor group (Fig. 1a). At a Bonferroni-adjusted *P*-value cutoff of 0.1, we obtained 7182 genomic regions with significant differential methylation between the tumor and non-tumor samples, with nearly two-third of those regions showing hypomethylation in the tumors. As expected, the differentially methylated regions (DMRs) were enriched in various genomic elements[19] (Fig. 1b). Notably, we found that 369 FANTOM5 enhancers[20] were differentially methylated, leading to a clear enrichment of the DMRs in enhancers (Fig. 1b).

As our HCC MBDCap-seq data, which provide resolution up to the length of each sequenced DNA fragment, suggest potential dysregulation of enhancers, we performed single-base resolution whole-genome bisulfite sequencing (WGBS) on three HCC tumors in order to identify enhancers with strong differential methylation at their core regions. After standard processing, we obtained an average of 17.6 aligned reads per CpG site, which concurred with the recommended coverage of WGBS[21]. Using normal liver from public WGBS data as a comparator, we identified enhancers with the strongest changes (average beta value change $\geq 0.1$ across $\geq 10$ CpG sites) in tumors (Fig. 1c), which exhibited highly significant hypomethylation ($P < 2.2e-16$; Fig. 1d). Among these DMEs, 854 were hypomethylated while only 40 were hypermethylated in tumors (Fig. 1e). Using the FANTOM5 database, we also identified the potential target genes of these DMEs and further integrated RNA-seq data of the same paired HCC samples. Consistent with the previous observations that enhancer hypo- and hypermethylation could lead to target gene up- and downregulation, respectively[14,15], we identified a list of 27 DMEs whose DNA methylation levels had strong inverse correlations with target gene expressions (Supplementary Table 1).

Based on the number of enhancers that target a particular gene and its potential functional significance, we selected a FANTOM5 enhancer region that is ~90-kb downstream to its target gene *C/EBPβ* (Fig. 1f), a key hepatocyte transcription factor for liver regeneration[22,23], for in-depth analysis. With multiple enhancers (3 out of 27) targeted *C/EBPβ* and a high eRNA-mRNA correlation of 0.863 across 808 FANTOM5 samples, the selected highly confident enhancer-target pair exhibited significant hypomethylation-associated gene upregulation in HCC tumors (Supplementary Table 1). To verify this potential enhancer, we performed nanoscale chromatin profiling[24] of the same HCC tissues used for WGBS. Analysis of the chromatin profiles at the central 1-kb region of the hypomethylated *C/EBPβ* enhancer revealed substantial enrichment of H3K27ac, a high H3K4me1 to H3K4me3 ratio but low H3K27me3 (Fig. 1f), representing an active enhancer in human HCCs[11,25].

**Hypomethylation of *C/EBPβ* enhancer relates to poor prognosis.** We next validated the WGBS findings by bisulfite pyrosequencing using 48 pairs of human HCC tumor/non-tumor tissues, which exhibited the methylation levels of the 13 CpG sites within the *C/EBPβ* enhancer (Fig. 1g, h). Consistent with the WGBS findings, the average *C/EBPβ* enhancer methylation levels of the tumors (~40%) were lower than the non-tumor tissues (~55%, $P < 0.005$; Fig. 1i). To investigate the relationship between *C/EBPβ* methylation and expression, quantitative RT-PCR (qRT-PCR) was performed using 33 pairs of the HCC cases of which high quality RNA samples were available. Compared to the non-tumor tissues, we demonstrated significantly higher *C/EBPβ* mRNA expression in tumors ($P < 0.005$; Fig. 1j), which was negatively and significantly correlated with the *C/EBPβ* enhancer methylation ($P < 0.01$; Fig. 1k). In contrast, the *C/EBPβ* promoter methylation levels were invariable among the tumor and

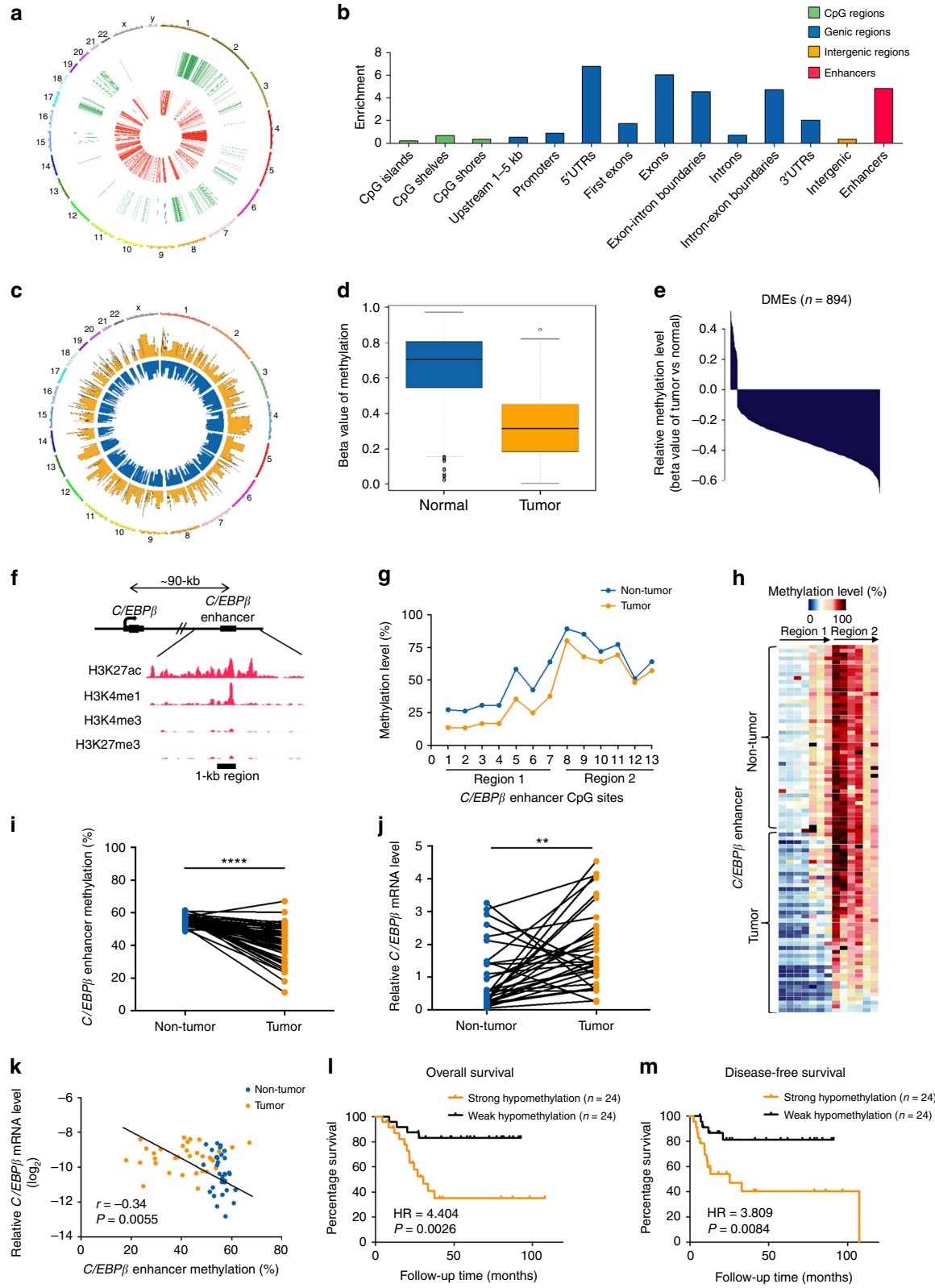

non-tumor tissues, exhibiting low methylation levels (< 2%) at the 26 CpGs encompassing the *C/EBPβ* transcription start site (TSS, −561 to + 458-bp; Supplementary Fig. 1a). We further examined our in-house and published whole-genome sequencing datasets[26,27]. *C/EBPβ* promoter mutations were observed in ~1% (4/363) of HCC patients, whereas no mutation could be found in *C/EBPβ* enhancer. Taken together, these data suggest that

enhancer hypomethylation is one of the major mechanisms for *C/EBPβ* over-expression in human HCC.

Based on the widespread tumorous *C/EBPβ* enhancer methylation levels (Fig. 1i), we further investigated whether enhancer methylation associates with the survival of HCC patients. Notably, Kaplan–Meier analysis revealed that HCC patients with strong tumorous hypomethylation (compared to the

**Fig. 1** HCC methylome analysis identifies a recurrent hypomethylated enhancer targeting a liver-enriched transcriptional factor C/EBPβ. **a** A circos plot showing genomic regions that are significantly hypomethylated (4528 regions, in red) or hypermethylated (2654 regions, in green) in the tumor group as compared to matched non-tumors based on MBDCap-seq data. **b** Enrichment of DMRs in annotated functional elements, defined as $(Lm_i/Lm)/(L_i/L)$, where $Lm_i$ is the total length of DMRs overlapping with the annotated functional elements of type $i$ and $Lm$ is the total length of all DMRs. $L_i$ is the length of the annotated functional elements of type $i$, and $L$ is the total length of all annotated functional elements combined. **c** Genome-wide methylation patterns of differentially methylated FANTOM5 enhancers in HCC tumor (orange) and normal liver (blue) tissues based on WGBS. **d** Enhancer methylation levels of HCC tumor and normal liver tissues. **e** Relative methylation levels of 894 DMEs between HCC tumor and normal liver tissues. **f** ChIP-seq tracks of H3K27ac, H3K4me1, H3K4me3, and H3K27me3 at the 1-kb C/EBPβ enhancer locus (chromosome 20: 48,900,221–48,901,229) in HCC tumor tissues. **g**–**i** Methylation levels of 13 CpG sites within the C/EBPβ enhancer in 48 pairs of HCC tumor and non-tumor tissues as determined by pyrosequencing. **j** qRT-PCR analysis of HCC tumor and non-tumor tissues (33 pairs). C/EBPβ mRNA levels were calculated by the $2^{-\Delta\Delta Ct}$ method using 18s rRNA as internal control, and are presented as fold-changes against the average value of the non-tumor group. **k** Correlation between C/EBPβ enhancer methylation and expression in 33 pairs of HCC tumor and non-tumor tissues. C/EBPβ mRNA levels are ΔCt values using 18s rRNA as internal control. **l**, **m** Kaplan–Meier survival analysis of 48 HCC patients according to their C/EBPβ hypomethylation statuses (relative methylation of tumor vs. non-tumor). Patients with strong hypomethylation (top 24) show poorer (**l**) overall and (**m**) disease-free survival rates than those with weak hypomethylation (bottom 24). Data are presented as mean ± SD. **P < 0.01; ****P < 0.0001 as calculated by Wilcoxon signed-rank test (**d**), paired two-tailed Student's t-test (**i**, **j**), Pearson correlation test (**k**) and Kaplan–Meier survival analysis (**l**, **m**)

corresponding non-tumor) significantly correlated with shorter overall (hazard ratio = 4.404, P < 0.005; Fig. 1l) and disease-free survival rates (hazard ratio = 3.809, P < 0.01; Fig. 1m). These results demonstrate that C/EBPβ enhancer hypomethylation was correlated with poorer prognosis of HCC patients.

**C/EBPβ enhancer hypomethylation activates C/EBPβ expression.** Given the causal role of eRNAs in transcriptional activation as suggested in some studies[11,28–30], we speculated that the methylation status of C/EBPβ enhancer controls eRNA expression for C/EBPβ gene regulation. First, we determined the presence of C/EBPβ enhancer-templated eRNAs by northern blot and qRT-PCR analyses. In both HepG2 liver cancer cells and immortalized LO2 liver cells, our data revealed production of a 3-kb polyadenylated RNA transcript from the sense strand, but not the antisense strand, of the C/EBPβ enhancer (Fig. 2a and Supplementary Fig. 2), which is consistent with the features of unidirectional eRNA[11]. Using a panel of eight HCC and liver cell lines, we next investigated whether C/EBPβ enhancer methylation controls eRNA expression by pyrosequencing and qRT-PCR. We found a significantly negative correlation between C/EBPβ enhancer methylation and eRNA expression (P < 0.005; Fig. 2b), while the C/EBPβ promoter methylation levels were invariably low (~10%; Supplementary Fig. 1b), similar to those observed in the primary HCC tissues. To test for the causative role, we treated PLC5 and SK-Hep1 liver cell lines that harbor high enhancer methylation levels ( > 50%) with 5-aza-2'-deoxycytidine (5-aza-dC). DNA demethylation was observed in C/EBPβ enhancer, but not promoter (Fig. 2c), which resulted in significant reactivation of C/EBPβ eRNA and mRNA in both lines (P < 0.05; Fig. 2d). To exclude potential influences by other hypomethylated sites upon 5-aza-dC treatment, we performed targeted DNA demethylation by a modified dCas9-TET1 hydroxylase fusion construct[31] and demonstrated that targeted demethylation of C/EBPβ enhancer increased C/EBPβ eRNA and mRNA expressions (P < 0.001; Supplementary Fig. 3a, b).

To further examine whether DNA methylation-regulated C/EBPβ eRNA is involved in transcriptional upregulation of C/EBPβ, we transfected two independent small interfering RNAs that target C/EBPβ eRNA (sieRNA) or a control sequence (siCtrl) into HepG2 and LO2 cells that highly express the eRNA (Fig. 2a, b). Intriguingly, knockdown of C/EBPβ eRNA in both lines reduced mRNA levels of C/EBPβ (Fig. 2e, f). We observed no change in the expression of the neighboring genes SMIM-25 and DPM1 located upstream and downstream of the C/EBPβ enhancer (Supplementary Figs. 2 and 3c), implying no off-target effect. In a complementary experiment,

we treated PLC5 and SK-Hep1 cells with 5-aza-dC or DMSO vehicle, followed by transfection with siCtrl or sieRNA. Notably, we found that C/EBPβ mRNA upregulation by DNA demethylation could be partially abrogated by knockdown of C/EBPβ eRNA (Fig. 2g, h). These results demonstrate that C/EBPβ enhancer hypomethylation is required for efficient C/EBPβ transcription via induction of eRNA. Furthermore, we found an elevation of C/EBPβ eRNA level in HCC tumor compared to non-tumor tissues, which exhibited significantly negative and positive correlations with C/EBPβ enhancer methylation (P < 0.05; Fig. 2i) and C/EBPβ mRNA levels (P < 0.01; Fig. 2j), respectively, thus verifying the clinical relevance of our findings.

**C/EBPβ feedback regulates its own enhancer activity.** We next speculated that C/EBPβ enhancer hypomethylation facilitates eRNA synthesis via enhanced co-occupancy of the transcriptional complex[10,32,33]. Indeed, DNA demethylation of C/EBPβ enhancer increased the occupancies of C/EBPβ, BRD4, H3K27ac, and RNA polymerase II (RNPII) at C/EBPβ enhancer in both PLC5 and SK-Hep1 cells (P < 0.05; Fig. 3a). Concordant with the enhancer co-regulatory function of C/EBPβ and BRD4[34], either siRNA-mediated downregulation of C/EBPβ (Fig. 3b) or treatment with the BET-bromodomain inhibitor JQ1 significantly reduced the C/EBPβ eRNA levels in HepG2 and LO2 cells (P < 0.01; Fig. 3c, d). These observations indicate that C/EBPβ co-localizes with BRD4 at its hypomethylated enhancer to promote eRNA production.

To further investigate the C/EBPβ-dependency of the enhancer activity, we cloned the 1-kb enhancer fragment, which contains a 10-bp consensus sequence (ATTGCACAAT) for C/EBP family members[35], downstream of the C/EBPβ promoter-driven luciferase gene (Fig. 3e). In both HepG2 and LO2 cells, C/EBPβ enhancer generated significantly higher luciferase activity compared to the C/EBPβ promoter alone (P < 0.01; Fig. 3f). Notably, deletion of C/EBPβ motif by site-directed mutagenesis partially abrogated the transcriptional activity of the enhancer (P < 0.05; Fig. 3e, f), which was consistent with the effect of C/EBPβ knockdown (P < 0.05; Fig. 3g). Enhancers control expression of genes over distance by DNA looping, which brings the distal regulatory elements into close proximity of their target gene promoters[36]. In concordance, the C/EBPβ enhancer has been shown to physically interact with the C/EBPβ promoter by previously generated Hi-C chromosome conformation capture data from two human cell lines (HeLa and K562)[37] (Supplementary Fig. 4), thus supporting long-range transcriptional regulation by C/EBPβ

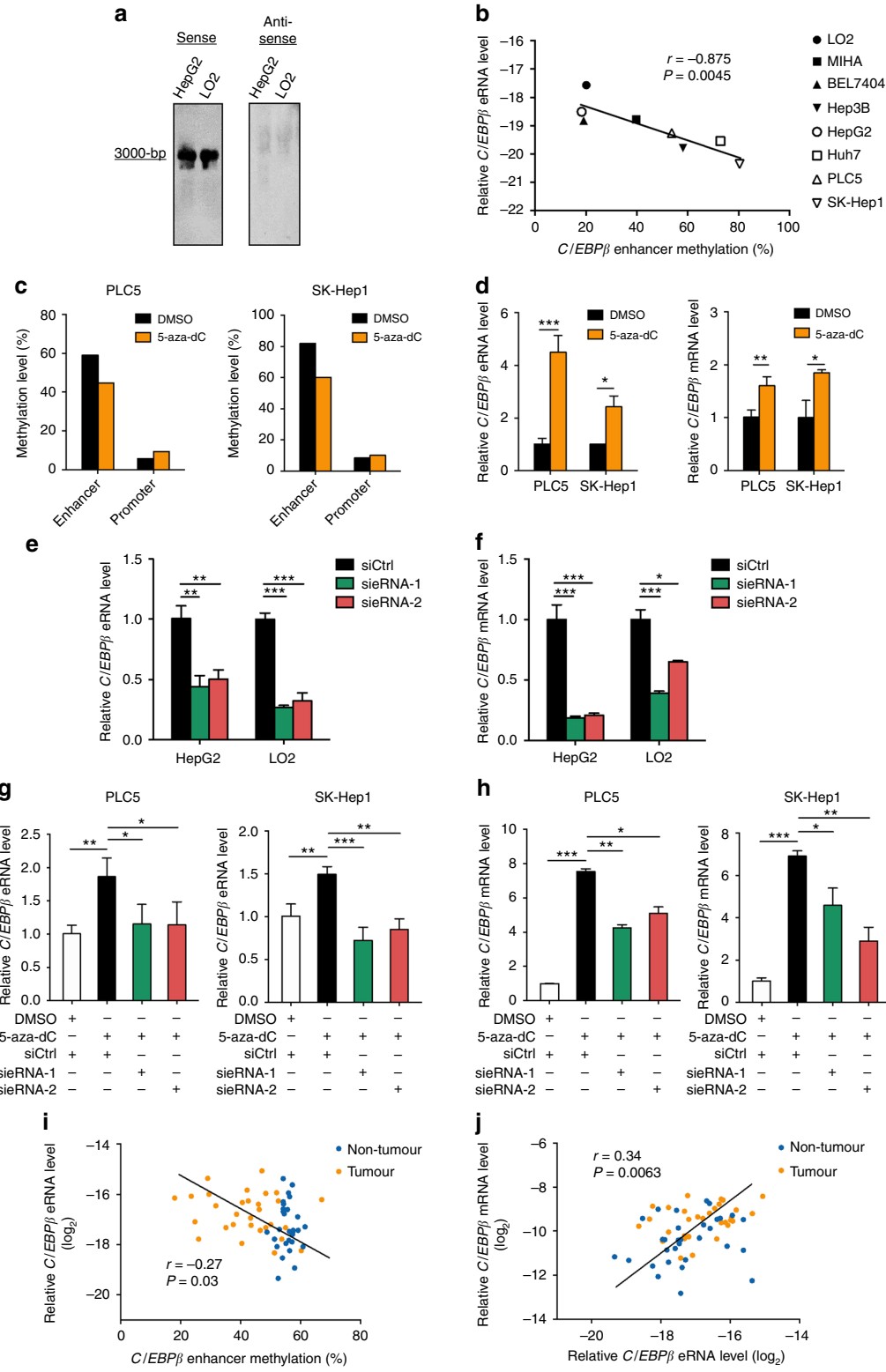

enhancer. Overall, these findings suggest that C/EBPβ feedback *trans*-activates its own enhancer via direct chromatin regulation.

***C/EBPβ enhancer deletion abrogates HCC tumorigenicity.*** To elucidate the biological and functional significance of C/EBPβ enhancer in tumor development, we generated monoallelic (*C/EBPβ enh*$^{+/-}$) and biallelic (*C/EBPβ enh*$^{-/-}$) deletion of

C/EBPβ enhancer via CRISPR/Cas9 in HepG2 and LO2 cells (Fig. 4a and Supplementary Fig. 5). Compared to the *C/EBPβ enh*$^{+/+}$ wild-type (WT) cells, *C/EBPβ enh*$^{+/-}$ and *C/EBPβ enh*$^{-/-}$ cells showed significantly reduced C/EBPβ eRNA and mRNA expressions in an allele-dependent manner (*P* < 0.01; Fig. 4b, c). Functionally, monoallelic and biallelic deletion of C/EBPβ enhancer caused a progressive reduction in cell growth (*P* < 0.05; Fig. 4d, e) and invasiveness (*P* < 0.05; Fig. 4f, g). Notably, C/EBPβ

**Fig. 2** *C/EBPβ* enhancer hypomethylation increases *C/EBPβ* expression via induction of eRNA. **a** Detection of *C/EBPβ* eRNA by northern blot using both sense and antisense probes. **b** Correlation between *C/EBPβ* enhancer methylation and eRNA expression in eight liver cell lines denoted with Pearson correlation coefficient. *C/EBPβ* eRNA levels are ΔCt values using 18s rRNA as internal control. **c** Pyrosequencing analysis of *C/EBPβ* promoter and enhancer methylation in liver cell lines treated with or without 5-aza-dC. **d** qRT-PCR analyses of *C/EBPβ* eRNA and mRNA in liver cell lines treated with or without 5-aza-dC. **e, f** Knockdown of *C/EBPβ* eRNA by two independent siRNAs in liver cells downregulated (**e**) *C/EBPβ* eRNA and (**f**) mRNA expressions. **g, h** Knockdown of *C/EBPβ* eRNA abrogated (**g**) *C/EBPβ* eRNA and (**h**) mRNA reactivation after 5-aza-dC treatment. **d-h** *C/EBPβ* eRNA/mRNA levels were calculated by the $2^{-\Delta\Delta Ct}$ method using 18s rRNA as internal control, and are presented as fold-changes against the average values of the respective control groups (DMSO/siCtrl). **i** Correlation between *C/EBPβ* enhancer methylation and eRNA levels in human HCC tissues denoted with Pearson correlation coefficient. **j** Correlation between *C/EBPβ* eRNA and mRNA levels in human HCC tissues denoted with Pearson correlation coefficient. **i, j** *C/EBPβ* eRNA/mRNA levels are ΔCt values using 18s rRNA as internal control. Data are presented as mean ± SD. *$P < 0.05$; **$P < 0.01$; ***$P < 0.001$ as calculated by Pearson correlation test (**b**, **i-j**), and unpaired two-tailed Student's *t*-test (**c-h**)

$enh^{+/-}$ and *C/EBPβ* $enh^{-/-}$ cells exhibited remarkable inhibition of tumor growth in xenograft models when compared with WT cells ($P < 0.005$; Fig. 4h, i). Indeed, *C/EBPβ* $enh^{-/-}$ HepG2 cells formed either no tumor (3 of 4) or very small tumor (1 of 4) within 4 weeks (Fig. 4h). We next investigated whether *C/EBPβ* eRNA exhibits similar pro-tumorigenic property. We found that downregulation of *C/EBPβ* eRNA in HepG2 and LO2 cells significantly reduced cell growth and invasiveness ($P < 0.01$; Supplementary Fig. 6a, b), and the extents of reduction appeared to be more than those in PLC5 and SK-Hep1 cells whose *C/EBPβ* enhancer was hypermethylated (Supplementary Fig. 6c, d). As expected, stable *C/EBPβ* knockdown phenocopied the effects of eRNA knockdown (Supplementary Fig. 6e, f). Taken together, these data suggest that *C/EBPβ* enhancer methylation may regulate cancer cell phenotypes via *C/EBPβ* eRNA/mRNA expression.

To examine the *C/EBPβ* methylation and expression changes leading to HCC, we took advantage of an HBV X protein (HBx) transgenic (TG) mouse model that spontaneously develops HCC tumors at old age[38–40]. Although highly conserved enhancers are much less common than promoters in mammalian species[41], high mouse-human conservation was observed in *C/ebpβ* enhancer, showing similar eRNA expression patterns and occupancy profiles for orthologous C/EBPβ and H3K27ac (Fig. 5a). For the two CpG sites located within the *C/ebpβ* enhancer, both were significantly hypomethylated in the livers of TG mice at the early stage of tumor development (4-month-old) when compared to the WT counterparts ($P < 0.05$; Fig. 5b). In contrast, no difference of methylation was observed in *C/ebpβ* promoter ($-645$ to $+312$-bp relative to TSS; Supplementary Fig. 7a). Moreover, the *C/ebpβ* eRNA ($P < 0.05$; Fig. 5c) and mRNA ($P < 0.05$; Fig. 5d) expressions were concordantly upregulated in TG compared to WT mice, and significantly correlated with each other at early and late (10-month-old) stages ($P < 0.05$; Supplementary Fig. 7b, c). In parallel, the C/EBPβ protein expressions were also significantly elevated ($P < 0.005$; Fig. 5e). These data suggest that *C/EBPβ* dysregulation by enhancer hypomethylation is a molecular event preceding the onset of HCC development in HBx TG mice.

***C/EBPβ* enhancer deletion remodels global enhancer activity**. Given the importance of C/EBPβ on global enhancer regulation[42], we speculated that *C/EBPβ* enhancer may influence genome-wide enhancer activity through its regulation of C/EBPβ. Indeed, deletion of *C/EBPβ* enhancer not only decreased the protein expression of C/EBPβ, but also the global level of H3K27ac in HepG2 cells (Fig. 6a). We next performed ChIP-seq analysis of C/EBPβ, BRD4 and H3K27ac in WT and *C/EBPβ* $enh^{-/-}$ HepG2 cells. A majority of C/EBPβ binding sites (84.2%) in WT cells showed reduced C/EBPβ and BRD4 co-occupancy in *C/EBPβ* $enh$ $^{-/-}$ cells (Fig. 6b), consistent with the role of C/EBPβ in chromatin targeting of BRD4[34]. Moreover, we identified reduction in

H3K27ac levels at > 30% enhancers (1800/5891) in *C/EBPβ* $enh$ $^{-/-}$ cells, whereas promoters showed significantly fewer changes ($P < 0.0001$; Fig. 6c). Notably, H3K27ac signals were diminished in *C/EBPβ* $enh^{-/-}$ cells at the majority of super-enhancer regions that lost C/EBPβ/BRD4 co-occupancy (Fig. 6d). Similarly, the H3K27ac levels of most C/EBPβ/BRD4-codepleted enhancers were also reduced in *C/EBPβ* $enh^{-/-}$ cells (Fig. 6e). As H3K27ac distinguishes active enhancers from inactive/poised enhancers[25], we investigated whether H3K27ac reduction affects target gene expressions in *C/EBPβ* $enh^{-/-}$ cells. Indeed, additional loss of H3K27ac at both C/EBPβ/BRD4-codepleted enhancers and super-enhancers correlated with reduced mRNA expressions of the nearest genes, as quantified by RNA-seq ($P < 0.0001$; Fig. 6f). Altogether, these findings indicate that *C/EBPβ* enhancer deletion impairs global enhancer activity via C/EBPβ/BRD4 dysregulation.

Gene ontology (GO) analyses[43] of the 805 super-enhancer- and 1817 enhancer-target genes whose activities were co-regulated by C/EBPβ and BRD4 (Fig. 6f) uncovered regulators of cancer pathways such as small GTPase-mediated signal transduction, and effectors of cell migration, proliferation, and vasculature development (Fig. 6g, h), consistent with the drastic inhibition of tumorigenicity in *C/EBPβ* $enh^{-/-}$ cells (Fig. 4). For example, *Ras like proto-oncogene B* (*RALB*)[44], a key mediator of metabolic and mitogenic pathways in HCC, was markedly downregulated in *C/EBPβ* $enh^{-/-}$ cells, which were associated with a notable decrease in the co-occupancy of C/EBPβ, BRD4, and H3K27ac at its super-enhancer region (Fig. 6i). Similarly, a pro-metastatic and pro-angiogenic factor over-expressed in human HCCs, namely *fibroblast growth factor receptor 2* (*FGFR2*)[45,46], was also deregulated (Fig. 6j). Notably, these HCC driver genes were not only suppressed in *C/EBPβ* $enh^{-/-}$ cells ($P < 0.05$; Fig. 6i, j), but also upregulated and correlated with *C/ebpβ* expression in the pre-malignant liver tissues of the HBx TG HCC model ($P < 0.05$; Supplementary Fig. 7d, e). Thus, our integrative genome-wide analysis implicates broad control of enhancer activity as the *C/EBPβ* enhancer function in the establishment and maintenance of HCC phenotypes.

## Discussion
Accumulating evidence has underscored aberrant enhancer-driven transcriptional programs as fundamental drivers of tumor formation and maintenance[13,47–49]. However, the mechanisms underlying enhancer dysregulation, especially from a DNA methylation viewpoint, remain unclear. Through comprehensive profiling of DNA methylation in primary tumors, matched non-tumor and normal liver tissues, we revealed global hypomethylation of transcriptional enhancers in human HCCs. Our study pinpointed an aberrantly-methylated enhancer with prognostic significance, forming a positive circuitry with its target gene to impart HCC hallmarks including proliferation, angiogenesis and invasion. C/EBPβ, over-expressed via its hypomethylated

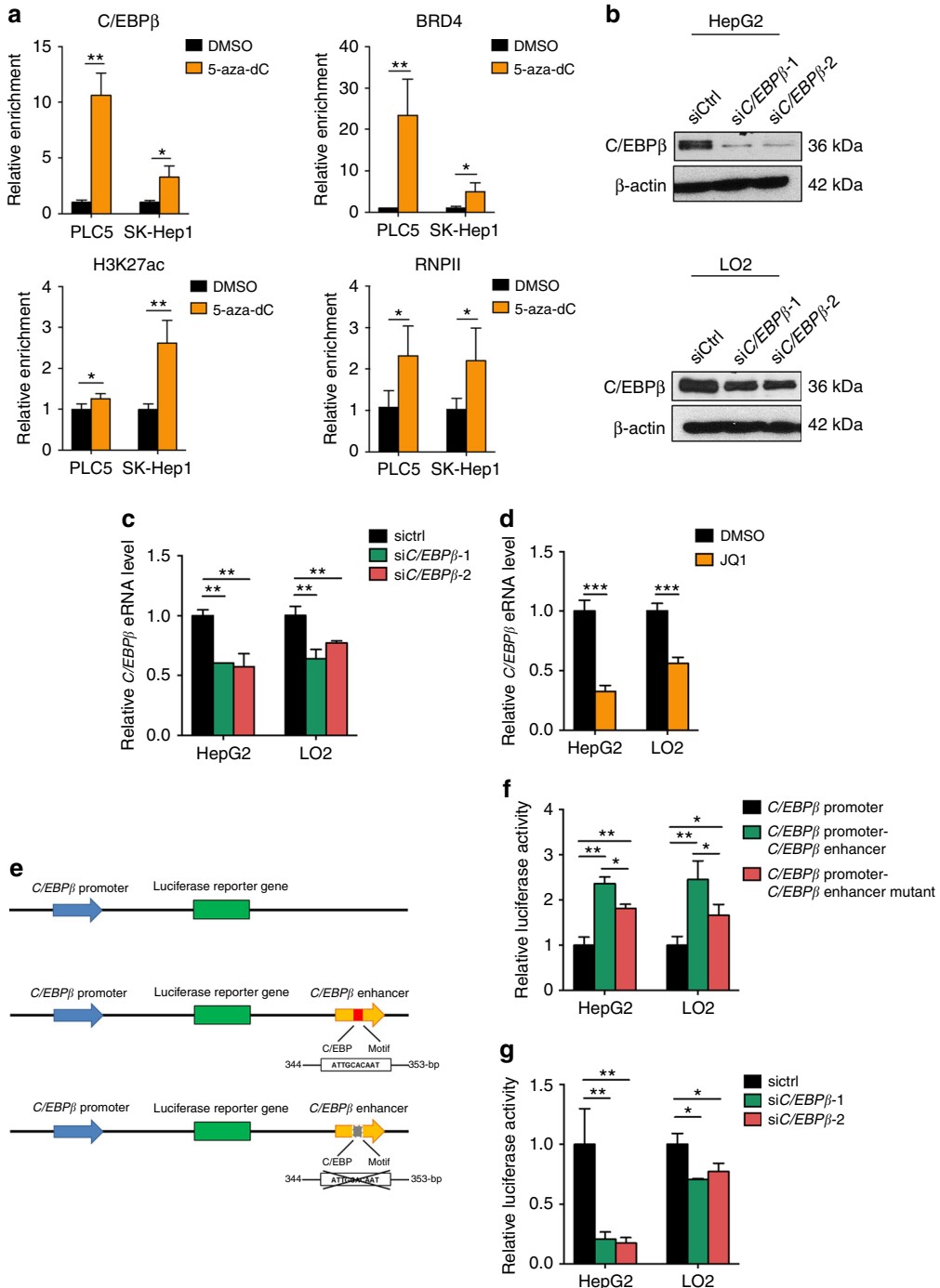

**Fig. 3** C/EBPβ auto-regulates its enhancer activity. **a** qChIP-PCR showed enrichment of C/EBPβ, BRD4, H3K27ac, and RNPII at *C/EBPβ* enhancer after 5-aza-dC treatment. **b** Western blot analysis of C/EBPβ level in liver cells upon siRNA-mediated knockdown. β-actin was used as loading control. **c**, **d** qRT-PCR analysis of *C/EBPβ* eRNA upon (**c**) *C/EBPβ* knockdown and (**d**) treatment with BRD4 inhibitor JQ1. *C/EBPβ* eRNA/mRNA levels were calculated by the $2^{-\Delta\Delta Ct}$ method using 18s rRNA as internal control, and are presented as fold-changes against the average values of the respective control groups (siCtrl/ DMSO). **e** Schematic diagrams of *C/EBPβ* promoter-, *C/EBPβ* promoter-*C/EBPβ* enhancer-, and *C/EBPβ* promoter-*C/EBPβ* enhancer mutant-luciferase reporter constructs. The *C/EBPβ* promoter (−285 to + 808-bp relative to TSS) was cloned to the PGL3 vector, in the absence or presence of the *C/EBPβ* enhancer with the WT or deleted C/EBP consensus sequence. **f** The relative luciferase activity in liver cells upon transfection with different reporter constructs. **g** The relative luciferase activity of the *C/EBPβ* enhancer construct upon siRNA-mediated knockdown of *C/EBPβ*. Data are presented as mean ± SD. *P < 0.05; **P < 0.01; ***P < 0.001 as calculated by unpaired two-tailed Student's t-test (**a**, **c**, **d**, **f**, **g**)

enhancer, has emerged as a crucial regulator of HCC tumorigenicity through genome-wide enhancer and super-enhancer remodeling. Given the well-established inheritance mechanism of DNA methylation, this study provides evidence for the heritability and causality of enhancer alterations in cancer development.

Our work has advanced the understanding of HCC methylome. To our knowledge, this is the first sequencing-based unbiased analysis that reveals a genome-wide enhancer hypomethylation pattern in human HCCs. While promoter hyper- and hypomethylation have been demonstrated in the development of

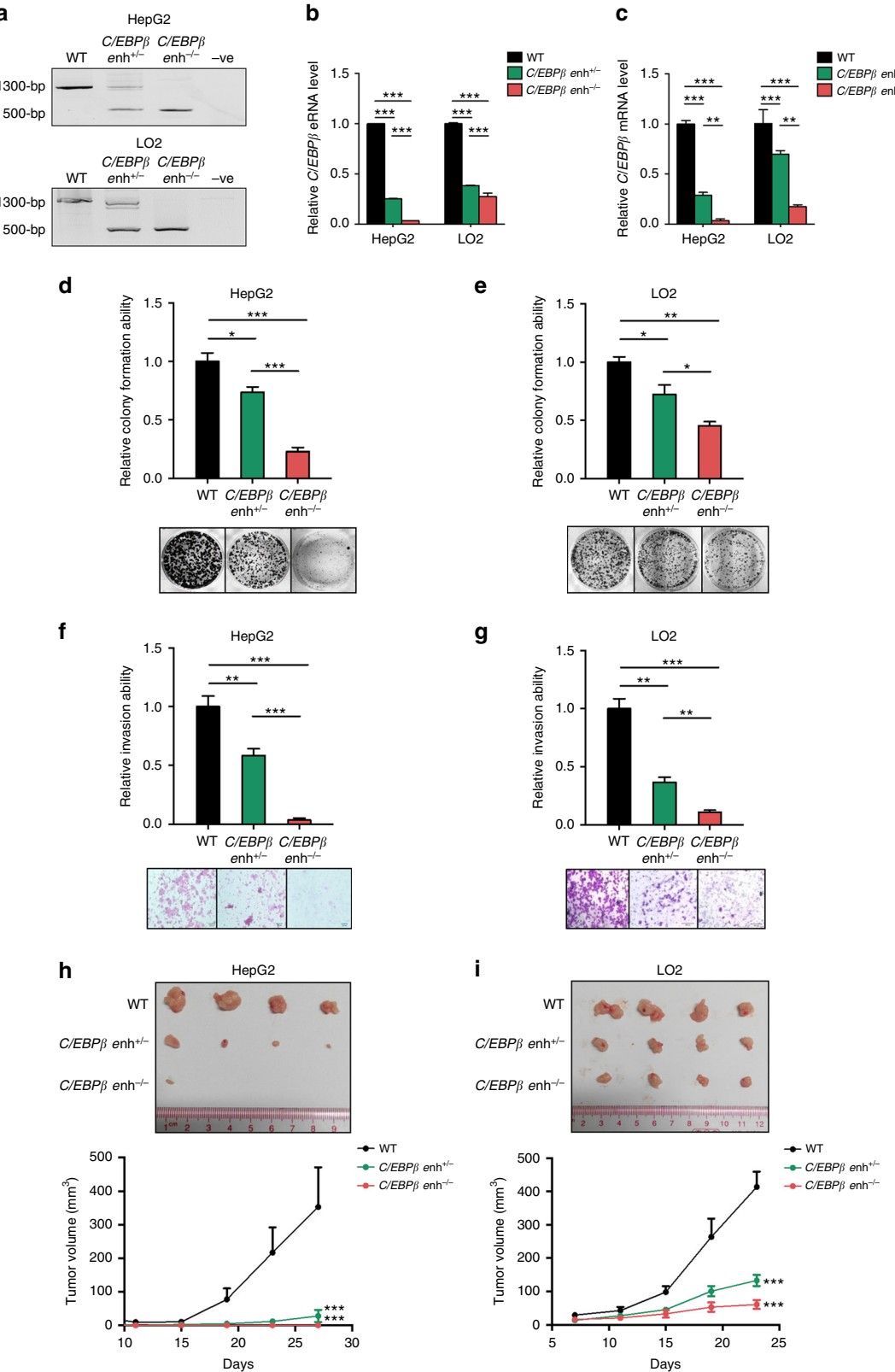

HCC[6–9], we found extensive reduction in methylation at the vast majority of DMEs identified by WGBS. Using nanoscale chromatin profiling in the same HCC tissues, we minimized ambiguity in enhancer localization and confirmed specific enhancer hypomethylation-associated *C/EBPβ* over-expression using bisulfite pyrosequencing and qRT-PCR in ~100 HCC tumor/non-

tumor tissues. Emerging data from leukemias[50], breast carcinomas[51], metastatic melanoma[16], and colorectal cancers[14] further support the notion that global enhancer hypomethylation is a common epigenetic feature of cancer development and progression. We and others have previously shown that the similarity of enhancer networks closely follows their cell and tissue

**Fig. 4** *C/EBPβ* enhancer functions to promote HCC tumorigenicity. **a** The monoallelic (*C/EBPβ enh*$^{+/-}$) and biallelic (*C/EBPβ enh*$^{-/-}$) deletions of *C/EBPβ* enhancer in HepG2 and LO2 liver cells via CRISPR/Cas9 were confirmed by PCR. **b**, **c** qRT-PCR analyses of **b** *C/EBPβ* eRNA and **c** *C/EBPβ* mRNA levels in WT, *C/EBPβ enh*$^{+/-}$ and *C/EBPβ enh*$^{-/-}$ cells. *C/EBPβ* eRNA/mRNA levels were calculated by the $2^{-\Delta\Delta Ct}$ method using 18s rRNA as internal control, and are presented as fold-changes against the average value of the WT group. **d**, **e** Cell proliferation of WT, *C/EBPβ enh*$^{+/-}$, and *C/EBPβ enh*$^{-/-}$ cells was determined by colony-formation assays. Representative images of focus formation are shown. **f**, **g** Cell invasion of WT, *C/EBPβ enh*$^{+/-}$, and *C/EBPβ enh*$^{-/-}$ cells was determined using Matrigel chambers. Representative images of crystal violet-stained invaded cells are shown. **h**, **i** Xenograft tumors formed by WT, *C/EBPβ enh*$^{+/-}$, and *C/EBPβ enh*$^{-/-}$ cells are shown in the images. Mean xenograft tumor volumes were plotted against days after injection. Data are presented as mean ± SD. *$P < 0.05$; **$P < 0.01$; ***$P < 0.001$ as calculated by unpaired two-tailed Student's *t*-test (**b–g**), and two-way ANOVA (**h**, **i**)

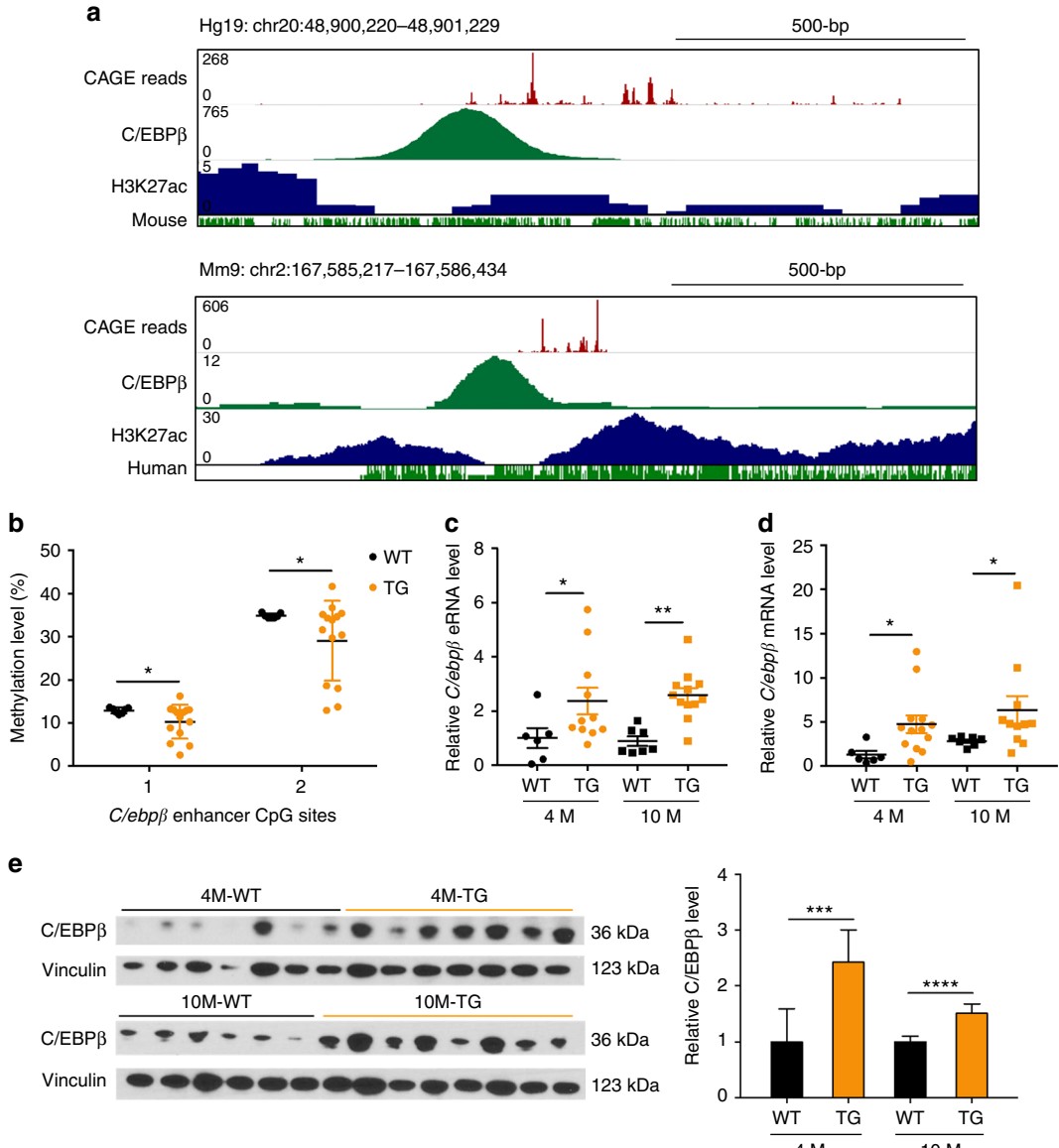

**Fig. 5** *C/ebpβ* enhancer hypomethylation associates with C/EBPβ over-expression in HBx TG mice. **a** Patterns of *C/EBPβ* eRNA (CAGE reads, FANTOM5), C/EBPβ and H3K27ac binding in the human HepG2 (GEO: GSM935493) and mouse hepatocyte genomes (GEO: GSM1854433). **b** Pyrosequencing analysis of *C/ebpβ* enhancer in liver tissues of 4-month-old WT and HBx TG mice (WT, $n = 7$; TG, $n = 14$). **c**, **d** qRT-PCR analyses of **c** *C/ebpβ* eRNA and **d** *C/ebpβ* mRNA levels in liver tissues of 4- and 10-month-old WT and HBx TG mice (4-month-old WT, $n = 7$; 4-month-old TG, $n = 12$; 10-month-old WT, $n = 7$; 10-month-old, $n = 12$). *C/ebpβ* eRNA/mRNA levels were calculated by the $2^{-\Delta\Delta Ct}$ method using 18s rRNA as internal control, and are presented as fold-changes against the average value of the 4-month-old WT group. **e** Western blot analysis of C/EBPβ protein level in liver tissues of 4- and 10-month-old WT and HBX TG mice (4-month-old WT, $n = 7$; 4-month-old TG, $n = 7$; 10-month-old WT, $n = 6$; 10-month-old, $n = 8$). Vinculin was used as loading control. The protein band intensities are quantified and shown on the right. Data are presented as mean ± SD. *$P < 0.05$; **$P < 0.01$; ***$P < 0.001$; ****$P < 0.0001$ as calculated by unpaired two-tailed Student's *t*-test (**b–e**)

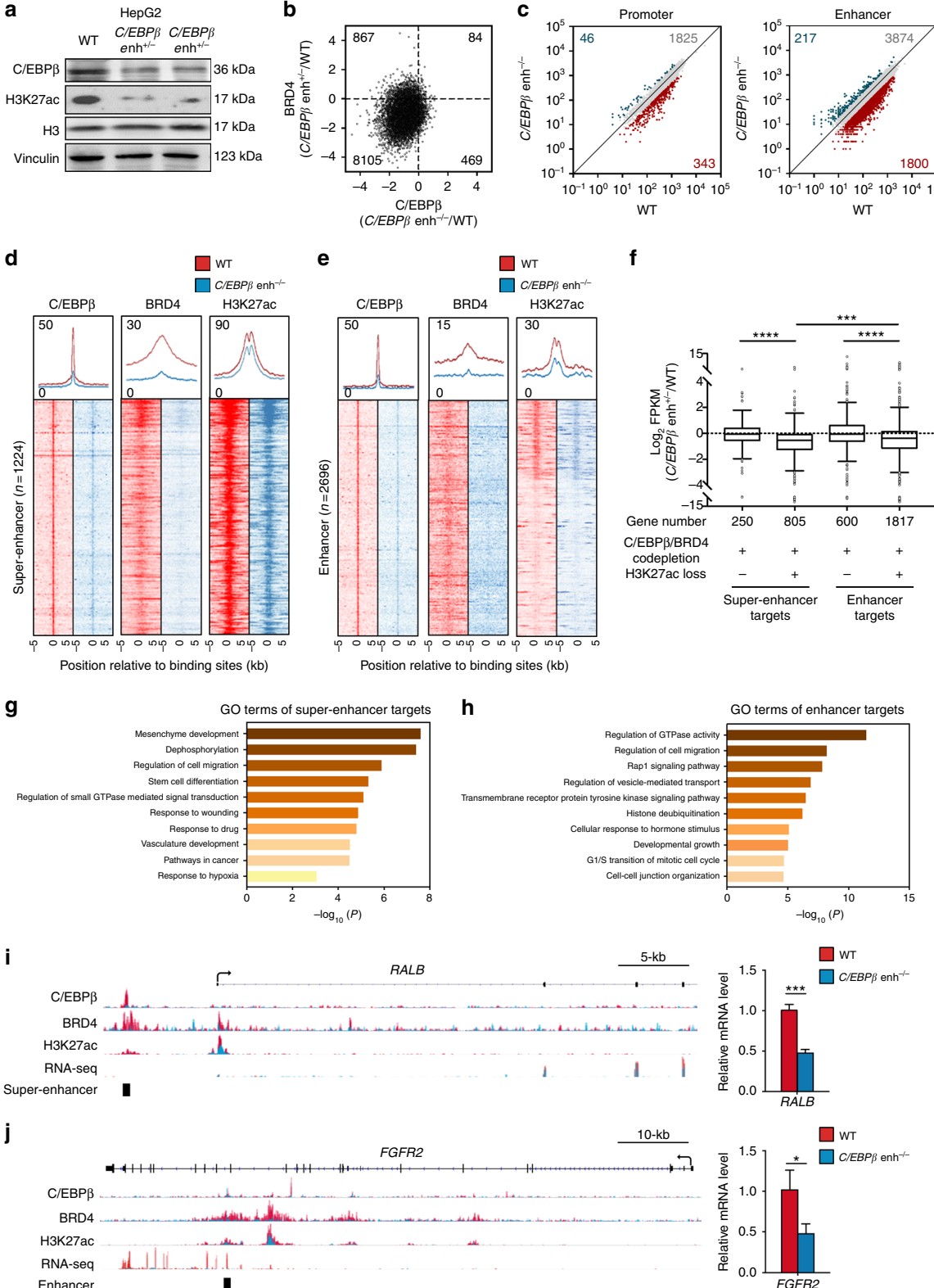

lineages[17,52]. Whether distinct enhancer methylation patterns can specify primary tumors according to their origins and stages remain to be elucidated in future studies.

Our enhancer methylation profile contains a wellspring of both well-validated and uncharacterized targets that may contribute to HCC tumorigenesis. Even though C/EBPβ has been previously reported to orchestrate liver regenerative[23] and glucose metabolic

programs[53], our study is the first to uncover its pro-tumorigenic properties in HCC. Other HCC-related genes such as the *SRC* tyrosine kinase[54] and the autophagy-related pro-survival gene *ATG7*[55] have been identified (Supplementary Table 1). New enhancer-hypomethylated and over-expressed targets revealed in this study include *IFNGR2* and *SLC45A4*, which could play important roles in HCC given their functions in regulating HBV

**Fig. 6** C/EBPβ enhancer deletion impairs genome-wide enhancer activity and chromatin regulation of driver oncogene expressions. **a** Western blot analysis of C/EBPβ and H3K27ac levels in C/EBPβ enh$^{+/-}$ and C/EBPβ enh$^{-/-}$ compared to WT HepG2 cells. H3 and vinculin were used as loading controls. **b** Fold change (log$_2$) in C/EBPβ and BRD4 ChIP-seq signals at C/EBPβ binding sites in C/EBPβ enh$^{-/-}$ relative to WT HepG2 cells. **c** H3K27ac levels in WT and C/EBPβ enh$^{-/-}$ cells in TSS-proximal promoter ( $+/-$ 2-kb) and TSS-distal enhancer ( >2-kb) enrichment regions (the numbers in red, green, and gray of each plot denote the regions with H3K27ac-loss ( <1/2×), -gain ( >2×), and stable in C/EBPβ enh$^{-/-}$ cells). **d, e** ChIP-seq profiles of C/EBPβ, BRD4, and H3K27ac in WT and C/EBPβ enh$^{-/-}$ cells around all C/EBPβ/BRD4-codepleted **d** super-enhancer and **e** enhancer sites. The average ChIP-seq signals from WT and C/EBPβ enh$^{-/-}$ cells are shown at the top. **f** Fold change (log$_2$) in gene expression between C/EBPβ enh$^{-/-}$ and WT cells for genes nearest to C/EBPβ/BRD4-codepleted enhancers or super-enhancers with or without H3K27ac loss. Boxes represent 1st, 2nd, and 3rd quartiles, and whiskers show 1.5 times the interquartile range below and above the 1st and 3rd quartiles, respectively. **g, h** GO analyses in **g** super-enhancer- and **h** enhancer-target genes using Metascape. The length of the bars represents the level of enrichment measured as a ratio between the number of genes overlapping an MSigDB gene set over the expected frequency if such overlaps were to occur at random. **i, j** C/EBPβ, BRD4, and H3K27ac ChIP-seq, RNA-seq tracks and enhancer/super-enhancer calls at the **i** RALB and **j** FGFR2 loci in WT and C/EBPβ enh$^{-/-}$ HepG2 cells. Gene expressions determined by qRT-PCR are also shown in right. The mRNA levels were calculated by the $2^{-\Delta\Delta Ct}$ method using 18s rRNA as internal control, and are presented as fold-changes against the average value of the WT group. Data are presented as mean ± SD. *P < 0.05; ***P < 0.001; ****P < 0.0001 as calculated by unpaired two-tailed Student's t-test (**f–j**)

---

viraemia[56] and redox homeostasis[57]. Whereas the distinct DNA hypermethylation profile in human HCCs is disproportionately enriched with the gain-of-function IDH1/2 mutations[9], the causes of enhancer hypomethylation warrant further investigation. One potential causative factor may be the HBx oncoprotein, which has been shown to induce demethylation of distal regulatory regions to facilitate HCC tumorigenesis[58]. Indeed the C/ebpβ enhancer regulatory network is recapitulated in an HBx TG HCC model, highlighting the importance of HCC risk factors in enhancer dysregulation during carcinogenesis.

Our data suggest that, mechanistically, reactivation of eRNA by C/EBPβ enhancer hypomethylation enables C/EBPβ transcription, which in turn binds to and trans-activates its own enhancer to form a self-reinforcing loop. Using Northern blot and strand-specific qRT-PCR analyses, we identified an approximately 3-kb eRNA that is unidirectionally transcribed for specific C/EBPβ gene regulation. Knockdown of this eRNA not only attenuates DNA demethylation-mediated C/EBPβ gene reactivation, but also inhibits HCC cell proliferation and invasion. As we also showed its upregulation and correlations with both C/EBPβ enhancer hypomethylation and mRNA expression in human HCCs, our findings concur with the functional significance of other tumor-specific eRNAs, such as androgen- and estrogen-dependent eRNAs in prostate[28] and breast cancers[29] that play indispensable roles in gene transcription programs[11].

One notable finding from this epigenomic study is the tumorigenic requirement of C/EBPβ enhancer in HCC. Deletion of this enhancer resulted in drastic abrogation in HCC tumorigenicity, which was associated with genome-wide codepletion of C/EBPβ and BRD4 occupancy and extensive dysregulation of gene expressions. Although C/EBPβ function is described at both promoters and enhancers[23,42], our findings implicate enhancers as the principal sites at which C/EBPβ/BRD4 complexes target to regulate gene activation in HCC cells. When C/EBPβ enhancer is absent, C/EBPβ/BRD4 is lost from thousands of enhancers that subsequently lose activity—showing reduced H3K27ac levels and expression of nearest genes. Notably, the activities of some super-enhancers, whose target genes are enriched in crucial HCC pathways, namely Ras, PI3K-Akt and MAPK signaling cascades, are also suppressed. For example, RALB is a small GTPase effector of RAS signaling that plays important roles in the proliferation, survival, and metastasis of a variety of human cancers[59]. C/EBPβ enhancer loss also impairs C/EBPβ/BRD4's enhancer control of signaling molecules (FGFR2, EGFR, ITGB1, SMAD3, TGFB2) and transcription regulators (HIF1A, RARA, FOXP1, YAP1, ZEB2) critical for angiogenesis and invasion. Given the concordant upregulation of the oncogene orthologs (Fgfr2, Hif1a, Ralb, Rara) in HBx TG mouse model, which is

correlated with C/EBPβ, our results suggest a paradigm of enhancer regulation of oncogenic cell signaling for further investigation. On the other hand, the differences in C/EBPβ enhancer methylation levels in clinical specimens and mouse tissues suggest that a fraction of the total HCC cells harbors C/EBPβ enhancer hypomethylation, which may reflect heterogeneity of the epigenome[60]. In concordance, modest methylation changes were also observed in the hypomethylated super-enhancers regulating the MYC and other oncogenes in the primary colon tumors in comparison with the matched normal colon mucosa[14]. It is thus conceivable that epigenetic heterogeneity can drive variable tumor-propagating potential, which could be fully delineated by single-cell epigenomic profiling[60].

The enhancer landscapes profiled in this study have implications beyond HCC. The oncogenic role of C/EBPβ, a liver-enriched transcription factor required for hepatocyte proliferation[22,23], is consistent with the other lineage-survival oncogenes, namely AR in prostate cancer[61], MITF in melanoma[62], GATA6 in lung adenocarcinoma[63] and OCA-B in diffuse large B cell lymphoma[48]. It is, therefore, conceivable that the lineage dependency or addiction[64] can be originated by genetic and/or epigenetic alterations such as enhancer hypomethylation. Targeting C/EBPβ by small-molecule inhibitors[65] may be therapeutically relevant to HCC and other human malignancies driven by alterations in C/EBPβ[66,67]. Like other cancers, HCC is characterized by global DNA hypomethylation, promoter hypermethylation, and more recently, hypomethylation associated with tumor-promoting gene upregulation[6,68]. Altogether with our discovery of aberrant enhancer hypomethylation that drives vicious positive feedback circuitry, cautions should be taken for the use of DNA hypomethylating agents as cancer therapeutics. The recent advancement in CRISPR/Cas9-mediated epigenome editing[31,69] may offer more specific treatment strategies via enhancer reprogramming.

## Methods

**Patients and integrative epigenomic analysis.** HCC patients who underwent hepatectomy at the Prince of Wales Hospital (Hong Kong) were included in this study (Supplementary Table 2). The specimens were processed immediately after surgery and snap-frozen in liquid nitrogen for DNA and RNA extraction. Informed consent was obtained from all human subjects, and the study protocol was approved by the Joint Chinese University of Hong Kong-New Territories East Cluster Clinical Research Ethics Committee. Our integrative epigenomic analysis is comprised of MBDCap-seq, WGBS, nanoscale chromatin profiling, and RNA-seq. The procedures and data analyses are provided in the Supplementary Information.

**Bisulfite pyrosequencing.** DNA samples were bisulfite-converted using EZ DNA Methylation™ Kit (Zymo). Primers targeting the DNA regions were designed by Qiagen PyroMark Assay Design SW 2.0. Taqgold DNA polymerase (Thermo-Fisher) was used for PCR amplification. A biotin-labeled primer was used for PCR

product purification by Sepharose beads according to manufacturer's protocols. The PCR product was sequenced by PyroMark MD Q96 MD System with sequencing primer. The level of CpG methylation was calculated by the signal to ratio of methylated cytosines to unmethylated cytosines. Non-CpG cytosine within the target regions was used as control to verify complete bisulfite conversion.

**Cell culture, 5-aza-dC/JQ1 treatment, and gene knockdown**. The eight immortalized human liver cell lines, LO2 (Cellosaurus, CVCL_6926), MIHA (Cellosaurus, CVCL_SA12), BEL7404 (Cellosaurus, CVCL_6568), Hep3B (ATCC, HB-8064), HepG2 (ATCC, HB-8065), Huh7 (JCRB, 0403), PLC5 (ATCC, CRL-8024), and SK-Hep1 (ATCC, HTB-52), were maintained in Dulbecco's modified Eagle's medium (Gibco) with 10% fetal bovine serum (HyClone). All cells were cultured at 37 °C in a humidified chamber containing 5% $CO_2$. For pharmacological DNA demethylation and BRD inhibition experiments, liver cells seeded in 6-well plate at 50% confluency overnight were treated with 10 μM of 5-aza-dC (Sigma–Aldrich) for 3 days and 2.5 μM JQ1 (Selleck) for 2 days, respectively. In all, 25 nM siRNA was transfected into cells for 48 h using HiPerfect (Qiagen). Two siRNAs targeting *C/EBPβ* were purchased from ThermoFisher. Two siRNAs targeting *C/EBPβ* eRNA were designed based on BLOCK-iT™ RNAi Designer. The sense and antisense sequences of si*C/EBPβ*−1 are GGGAGGUAAAUAUA-GUGCCUGTT and CAGGCACUAUAUUACCUCCTT. The sense and antisense sequences of si*C/EBPβ*−2# are GGUAAAUAUAGUGCCUGCUCTT and GAG-CAGGCACUAUAUUACCTT. The short-hairpin RNA (shRNA) vector targeting *C/EBPβ* was purchased from GenePharma. *C/EBPβ* expression vector was purchased from FulenGen. CRISPR/Cas9 blank vector was purchased from Addgene. Construction of CRISPR-*C/EBPβ* eRNA-sgRNA vectors was performed following the protocol by Addgene.

**Targeted DNA demethylation**. The pPlatTET-gRNA2 vector for dCas9-GCN4 and scFv-TET1CD-GFP fusion protein expression was obtained from Addgene (#82559)[31]. A single guide RNA (sgRNA) expression vector was modified from MLM3636 (Addgene #43860), in which 20-bp sgRNAs designed by CRISPR/Cas9 Target Online Predictor (CCTop)[70] were cloned. The sgRNA sequences for a control region[69] and *C/EBPβ* enhancer are 5′-CCCCCGGGGGAAAAATTTTT-3′ and 5′-CACACACACAGGGCCACCGA-3′, respectively. The pPlatTET-gRNA2 and sgRNA-expressing vectors were co-transfected into SK-Hep1 cells by jet-PRIME from Polyplus Transfection according to the manufacturer's instructions. After 48 h, transfected cells were flow-sorted to isolate GFP-positive cells, followed by cell expansion for pyrosequencing and qRT-PCR analyses.

**Luciferase assay and site-directed mutagenesis**. The *C/EBPβ* promoter and enhancer regions were cloned into pGL3–basic vector (Promega) to generate the luciferase reporters. The *C/EBPβ* enhancer luciferase plasmid with deletion of C/EBP binding motif (ATTGCACAAT) was generated by the QuikChange II Site-Directed Mutagenesis Kit (Stratagene). All plasmids were verified by DNA sequencing.

**RNA extraction and qRT-PCR**. RNA was extracted using RNA extraction kit (Fastagen) or Trizol and quantified by NanoDrop ND-2000 (NanoDrop Technologies). For reverse transcription, 2 μg of RNA was first treated with DNASE (Invitrogen). PrimeScript RT Master Mix (Takara) was then used for cDNA generation. For quantitative PCR analysis, SYBR Green PCR Master Mix (Takara) was used. The target genes were amplified in QS7 or 7500 Fast Real-Time PCR machine (Applied Biosystems). The experiments were replicated three times in two independent experiments. The primers are listed in Supplementary Table 3.

**Western blot**. The primary antibodies for western blotting are CEBPB (sc-150, Santa Cruz Biotechnology, 1:1000), β-actin (8H10D10, Cell Signaling Technology, 1:10,000), Vinculin (sc-25336, Santa Cruz Biotechnology, 1:1000), H3K27ac (39133, Active Motif, 1:1000), and H3 (4499, Cell Signaling Technology, 1:1000). Cellular proteins were extracted in lysis buffer (50 mM Tris-HCl, pH 7.5, 150 mM NaCl, 1% NP-40, 0.5% Na-deoxycholate) with protease inhibitors (Roche), and their concentrations were measured by DC Protein Assay (Bio-Rad). Protein samples were loaded into 8–12% sodium dodecyl sulfate polyacrylamide gel for electrophoresis and transferred to nitrocellulose membrane (Bio-Rad). Primary and secondary antibodies were sequentially added, and the target proteins were detected by Enhanced Chemiluminescence (GE Healthcare Life Sciences). β-actin or vinculin served as loading control. Uncropped western blots are shown in Supplementary Fig. 8.

**Cell proliferation, colony formation, and cell invasion**. For measuring cell proliferation, $10^3$ cells were seeded in individual wells of 96-well plate, and CellTiter Proliferation Assay kit (Promega) was used according to manufacturer's instructions. For measuring colony formation, cells seeded on a 6-well plate at 50–80% confluency were transiently transfected with plasmids. After 2 days, the cells were cultured in antibiotic-containing medium for 2–3 weeks. The resistant colonies were stained with 0.2% crystal violet and counted in three independent experiments. For cell invasion assay, BD BioCoat Matrigel invasion chambers (BD Biosciences) was used. The cells were fixed and stained with crystal violet, followed by counting. The experiments were replicated three times in two independent experiments.

**Animal experiments**. The experimental use of all mice was approved by the Animal Experimentation Ethics Committee of the Chinese University of Hong Kong. All mice received humane care according to the criteria outlined in the Guide for the Care and Use of Laboratory Animals (NIH). The strain of HBx TG mice was fixed to C57BL/6 by backcrossing with the C57BL/6 strain for more than 20 generations[37]. Male HBx TG and WT mice were sacrificed at either 4 or 10 months of age, and the liver tissues were excised and snap-frozen. For xenograft assay, $5 \times 10^6$ cells were subcutaneously injected into the left and right flanks of nude mice. Tumor size was measured using a caliper, and the tumor volume was calculated as $0.5 \times l \times w^2$, with $l$ indicating length and $w$ indicating width. The mice were euthanized at 5–6 weeks, and the tumors were excised and snap-frozen.

**Statistical analysis**. GraphPad Prism 5 (GraphPad Software) was used for statistical analysis. The difference between two groups was calculated by independent Student's $t$-test or Wilcoxon signed-rank test. The correlation among methylation and eRNA/mRNA expression was analyzed using Pearson correlation test. Chi-square test with Yates correction was used to determine whether distributions of H3K27ac levels differ between promoters and enhancers. Kaplan–Meier survival analysis was used to determine the overall and disease-free survival rates, which were calculated from the date of curative surgery to death, HCC recurrence or the last follow-up; the differences were compared by log-rank Mantel–Cox test.

**Reporting summary**. Further information on experimental design is available in the Nature Research Reporting Summary linked to this article.

## Data availability
All data supporting the findings of this study are available within the article and Supplementary Information, or from the corresponding author upon request. All ChIP-seq and RNA-seq data generated in this study are deposited in the Gene Expression Omnibus (GEO) database under the accession numbers "GSE123097", "GSE123098", and "GSE123099." Public sequencing datasets used in this study are: patterns of *C/EBPβ* eRNA (CAGE reads, 'FANTOM5 [http://fantom.gsc.riken.jp/5/data]'), and C/EBPβ and H3K27ac binding in the human HepG2 genome (GEO: "GSM935493") and mouse hepatocyte genome (GEO: "GSM1854433"). A reporting summary for this article is available as a Supplementary Information file.

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

## Acknowledgements

This project is supported by the University Grants Committee through the Collaborative Research Fund C4017–14G and the Focused Innovations Scheme-Scheme B 1907309 from the Chinese University of Hong Kong (CUHK). Alfred Cheng is supported by funding from the Young Researcher Award, CUHK.

## Author contributions

Study concept and design: K.Y.Y., A.S.C., K.F.T.; acquisition of data: L.X., F.W., L.X., O.K.C., W.K., L.L.M.S., C.Y.L., R.W.L., J.Z., G.H., C.M.W., J.L., J.L.C., W.H.H.; analysis and interpretation of data: L.X., F.W., Q.W., L.X., O.K.C., W.K., M.T.M., K.H.Y., S.D.L., G.H., C.M.W., J.L., T.H.H., K.Y.Y., A.S.C., K.F.T.; acquisition of patient specimens: Y.S.C., P.B.L.; drafting of the manuscript and preparation of figures: L.X., O.K.C.,

M.T.M., A.S.C.; critical revision of the manuscript: T.H.H., K.Y.Y., A.S.C., K.F.T.; obtained funding: N.W., K.Y.Y., A.S.C., K.F.T.; administrative, technical or other material support: Z.Y., D.Y.Y., B.F., P.T., N.W., M.W.C., T.H.H.; study supervision: K.Y.Y., A.S.C., K.F.T.

## Additional information

**Competing interests:** The authors declare no competing interests.

