## [Peer Review File · Nature Communications]

Reviewers' comments:

Reviewer #1 (Remarks to the Author):

In this manuscript, Xiong et al found that C/EBPbeta enhancer was hypomethylated in human and HBx-transgenic (TG) mouse hepatocellular carcinomas (HCCs), and demonstrate that the hypomethylation leads to increased H3K27ac, enhancer RNA expression, and increased C/EBPbeta expression. They also show that C/EBPbeta expression augments the enhancer activity, that a positive feedback loop is formed, and that deletion of C/EBPbeta enhancer dramatically decreases tumorigenicity of HCC cell lines.

The study convincingly shows the importance of C/EBPbeta enhancer hypomethylation and formation of a vicious positive feedback loop. Involvement of enhancer alterations in cancer is now attracting great attention. Although there are multiple studies, as cited, that show the importance of super-enhancers, whose inheritance mechanisms are still unclear, this study will be one of the first studies that show heritable and causal alterations of enhancers. Experiments in the study were carefully conducted using human and mouse systems and multiple cell lines/siRNA for a single experiment.

Major Comments:

1. Fig. 1. The selection process of C/EBPbeta enhancer as a top candidate may appear as a result of genome-wide screening, but is still unclear. From the 27 differentially methylated enhancers (DMEs), the authors selected C/EBPbeta enhancer based upon its gene function (line marks 118-120). This gives an impression that the selection was arbitrary. In addition, the role of C/EBPbeta enhancer hypomethylation in HCC development is expected to be highly variable among the 33 HCCs analyzed. Without information on mutations of known HCC driver genes/pathways, the selection of C/EBPbeta enhancer does not appear to be non-biased. This reviewer feels that, even if the authors focused on C/EBPbeta enhancer from the beginning based upon its biological knowledge, there would be no problem.

2. Fig. 1g and 1h. The difference in methylation levels between tumor and non-tumor tissues was 10-15%. Since methylation status of one allele should be plus or minus, the data means that demethylation of one allele took place in 20-30% of cells in tumor tissues or that of both alleles took place in 10-15% of cells in tumor tissues, supposing no allelic imbalances. The fractions appear to be a fraction of the total HCC cells. How do the authors explain that hypomethylation in a fraction of HCC cells drives the tumorigenicity of the entire tumor tissues?

3. Fig. 2i. Correlation analysis using two different groups of samples (non-tumor and tumor samples) may lead to false association between methylation and eRNA expression.

4. Fig. 5b. The degree of hypomethylation in the TG mice is highly variable. The authors used TG mice at 4 months (male?). In ref #37, although microscopic lesions were still absent at this age, such lesions were identifiable at 6 months. This suggests molecular changes have already started at 4 months to various degrees depending upon individuals, and this may explain the high variety of the hypomethylation at 6 months. The authors may want to correlate the molecular changes in Sup Fig 7d and the C/EBPbeta enhancer hypomethylation.

5. Fig. 6. The authors describe that C/EBPbeta enhancer affected global enhancer activity. This may be true, but it is not clear why they can conclude the changes observed were due to C/EBPbeta enhancer, not due to C/EBPbeta itself. Both C/EBPbeta enhancer deletion and eRNA/mRNA knock-down down-regulate C/EBPbeta, and the former is likely to have a stronger effect. If the authors want to keep the conclusion of C/EBPbeta enhancer, not due to C/EBPbeta itself, they need to compare the C/EBPbeta expression in Fig. 6a by adding eRNA/mRNA knock-down cells.

Minor Comments:

1. This study rings an alarm on the use of DNA hypomethylating agents for cancers in which hypomethylation is a driving vicious positive feedback loop. The authors may want to discuss the issue.

2. Throughout the manuscript. The authors use "relative GENE expression" for quantitative RT-PCR (for example, Fig. 1j, and 2b-2h). The meaning of "relative" seems to be different depending upon panels. For 1j and 2b, it appears that delta-Ct values are displayed while the expression levels were normalized to a specific sample in Fig. 2c-2h. The meaning needs to be clarified.

3. Fig. 7. The importance of this Fig. does not appear as high as that of the other Figs, and the authors may want to consider merging this one with Fig. 6 and moving some into a Supplementary Figs.

4. Supplementary Fig. 4. Vertical strips in red and green are misleading. The authors should simply mark the positions of the promoter and enhancer in the genome structure at the top.

5. Supplementary Fig. 6a. The degrees of eRNA knock-down need to be analyzed by qRT-PCR.
6. Description on the HBx-TG mice is lacking in the "Methods" or "Supplementary Methods" section.
7. Page 7, line mark 170. "5-aza-2-deoxycytidine (5'aza-dC)" -> "5-aza-2'-deoxycytidine (5-aza-dC)"
8. Page 7, line mark 180, and other places. "5'-aa-dC" -> "5-aza-dC"
9. Page 10, line marks 287-288. The authors may want to clarify the phrase "mechanisms underlying dysregulation remain unclear" by adding "from a DNA methylation viewpoint", and noting that the inheritance mechanism of DNA methylation is well established.

Reviewer #2 (Remarks to the Author):

Remarks to the author

This study can significantly contribute to uncover tumorigenesis of HCC. I sustain your enforcement. However I think your paper has some remediation points and propose following amendment points.

Major

#1. You demonstrated 5-aza-2dC treatment to make hypomethylation at C/EBP β enhancer lesion and suggested that C/EBP β enhancer hypomethylation is related to eRNA reactivation and expression. However, 5-aza-2dC induces genome wide hypomethylation, not specific at C/EBP β enhancer. Therefore, in the results, we are unable to disregard influences of other sites hypomethylation. I will give you a suggestion that you should make C/EBP β enhancer specific hypomethylation using epigenetic editing technology such as CRISPR/CAS9 system (DOI: 10.1002/anie.201601708, DOI: 10.1038/nbt.3658 etc.). If you make this specific hypomethylation model, you can make sure the

result that C/EBP β enhancer hypomethylation is related to HCC carcinogenesis. This model although help you to figure out the specific enhancer hypomethylation-associated C/EBP β over-expression when you demonstrate exhausted analysis of expression variation of eRNA or mRNA, changes of methylation profiles between before C/EBP β over-expression and after it.

#2. To indicate relationship between C/EBP β enhancer hypomethylation and eRNA and mRNA expression, you should do eRNA KD examination using both methylation rich cell lines and poor cell lines, and comparative investigation.

#3. At HBx TG mouse examination, your results only suggest the relationship between HBx TG and trends of C/ebp β hypomethylation, evaluation of eRNA, mRNA and protein levels. These results are feeble to suggest that C/EBP β dysregulation by enhancer hypomethylation occurs early in the carcinogenic cascade and plays a crucial role for HCC tumorigenicity.

Minor

- Introduction

#1. At introduction, I hope you demonstrate the reason you focused on C/EBP β , or preliminary data previously.

- Result

#2. At eRNA knock down experiment using siC/EBP β 1 and 2, do you state flatly that mRNA downregulation is not involved in off-target effect?

#3. At figure l,m, you should clarify standards of cutoff values by which you classified HCC patients into strong group and weak group.

- Methods

#4. You should clarify information of patient's characters, especially background of liver diseases. Readers are concerned about result when you classify patients according to causes of HCC such as HBV infection, HCV infection and NAFLD.

- figures

#5. At figure3c, you should correct siRNA name such as following.

siC/EBB β -1→siC/EBP β 1

Reviewer 1

Ref 1.1. Selection process of C/EBPbeta enhancer

Reviewer Comment	In this manuscript, Xiong et al found that C/EBPbeta enhancer was hypomethylated in human and HBx-transgenic (TG) mouse hepatocellular carcinomas (HCCs), and demonstrate that the hypomethylation leads to increased H3K27ac, enhancer RNA expression, and increased C/EBPbeta expression. They also show that C/EBPbeta expression augments the enhancer activity, that a positive feedback loop is formed, and that deletion of C/EBPbeta enhancer dramatically decreases tumorigenicity of HCC cell lines. The study convincingly shows the importance of C/EBPbeta enhancer hypomethylation and formation of a vicious positive feedback loop. Involvement of enhancer alterations in cancer is now attracting great attention. Although there are multiple studies, as cited, that show the importance of super-enhancers, whose inheritance mechanisms are still unclear, this study will be one of the first studies that show heritable and causal alterations of enhancers. Experiments in the study were carefully conducted using human and mouse systems and multiple cell lines/siRNA for a single experiment. Major Comments: 1. Fig. 1. The selection process of C/EBPbeta enhancer as a top candidate may appear as a result of genome-wide screening, but is still unclear. From the 27 differentially methylated enhancers (DMEs), the authors selected C/EBPbeta enhancer based upon its gene function (line marks 118-120). This gives an impression that the selection was arbitrary. In addition, the role of C/EBPbeta enhancer hypomethylation in HCC development is expected to be highly variable among the 33 HCCs analyzed. Without information on mutations of known HCC driver genes/pathways, the selection of C/EBPbeta enhancer does not appear to be non-biased. This reviewer feels that, even if the authors focused on C/EBPbeta enhancer from the beginning based upon its biological knowledge, there would be no problem.
Author Response	We fully agree with the reviewer and acknowledge the fact that the role of C/EBPbeta enhancer hypomethylation in HCC development would be highly variable among the HCCs analyzed in consistent with tumor heterogeneity (see also our response to Ref 1.2 below). As this enhancer has not been studied before, we think that it may be better to illustrate how this candidate was selected. In addition to gene function, C/EBPbeta enhancer was selected based upon another important reason not mentioned in the original manuscript, namely the number of enhancers that target a particular gene. Having multiple enhancers increase our confidence that the target genes were affected by differential enhancer methylation. C/EBPbeta had the largest number of enhancers (3 of them) on the list of 27 DMEs. The only other gene with 3 enhancers on the list is LITAF. The C/EBPbeta enhancer we selected had the highest eRNA-mRNA correlation (0.863) based on FANTOM5 among all the enhancers on the list coming from a gene with multiple enhancers.

Excerpt from revised manuscript	(P.5:) Based on the number of enhancers that target a particular gene and its potential functional significance, we selected a FANTOM5 enhancer region that is ~90-kb downstream to its target gene C/EBPβ (Fig. 1f), a key hepatocyte transcription factor for liver regeneration^{22,23}, for in-depth analysis. With multiple enhancers (3 out of 27) targeted C/EBPβ and a high eRNA-mRNA correlation of 0.863 across 808 FANTOM5 samples, the selected highly-confident enhancer-target pair exhibited significant hypomethylation-associated gene up-regulation in HCC tumors (Supplementary Table 1).
---

Ref 1.2. Difference in methylation levels between tumor and non-tumor tissues

Reviewer Comment	2. Fig. 1g and 1h. The difference in methylation levels between tumor and non-tumor tissues was 10-15%. Since methylation status of one allele should be plus or minus, the data means that demethylation of one allele took place in 20-30% of cells in tumor tissues or that of both alleles took place in 10-15 % of cells in tumor tissues, supposing no allelic imbalances. The fractions appear to be a fraction of the total HCC cells. How do the authors explain that hypomethylation in a fraction of HCC cells drives the tumorigenicity of the entire tumor tissues?
Author Response	The difference in C/EBPβ enhancer methylation levels between tumor and non-tumor tissues may reflect tumor heterogeneity, in which the bulk tumor includes a diverse collection of tumor cells harboring distinct molecular signatures with different phenotypic traits (Mazor et al. 2016). The composition of non-cancer cells in tumor microenvironment, although a minority compared to HCC cells, may also contribute to the modest methylation changes. Notably, tumor cell heterogeneity was also observed in the hypomethylated super-enhancer regulating the MYC oncogene, which was significantly associated with an increase in gene expression in the primary colon tumors in comparison with the matched normal colon mucosa (Heyn et al. 2016). It is thus conceivable that intratumoral heterogeneity of the epigenome can drive variable tumor-propagating potential, which could be fully delineated by single-cell epigenomic profiling (Mazor et al. 2016).
Excerpt from revised manuscript	(P.11-12:) On the other hand, the differences in C/EBPβ enhancer methylation levels in clinical specimens and mouse tissues suggest that a fraction of the total HCC cells harbors C/EBPβ enhancer hypomethylation, which may reflect heterogeneity of the epigenome⁶⁰. In concordance, modest methylation changes were also observed in the hypomethylated super-enhancers regulating the MYC and other oncogenes in the primary colon tumors in comparison with the matched normal colon mucosa¹⁴. It is thus conceivable that epigenetic heterogeneity can drive variable tumor-propagating potential, which could be fully delineated by single-cell epigenomic profiling⁶⁰.

Ref 1.3. Correlation analysis of non-tumor and tumor samples

Reviewer Comment	3. Fig. 2i. Correlation analysis using two different groups of samples (non-tumor and tumor samples) may lead to false association between methylation and eRNA expression.
------------------	--

Author Response	We have analyzed the correlations between methylation and eRNA expression in non-tumor and tumor samples separately, which unfortunately failed to show significance. Nevertheless, we observed a trend of inverse relationship between methylation and eRNA expression in tumor tissues, which may achieve significance with larger sample size. With reference to the association graphs of expression and DNA methylation levels at hypomethylated super-enhancers using both cancer and normal samples (Heyn et al. 2016), in the revised manuscript we continue to use combined data for correlation analysis.
-----------------	---

Ref 1.4. Correlation between *C/EBPbeta* enhancer hypomethylation and molecular changes

Reviewer Comment	4. Fig. 5b. The degree of hypomethylation in the TG mice is highly variable. The authors used TG mice at 4 months (male?). In ref #37, although microscopic lesions were still absent at this age, such lesions were identifiable at 6 months. This suggests molecular changes have already started at 4 months to various degrees depending upon individuals, and this may explain the high variety of the hypomethylation at 6 months. The authors may want to correlate the molecular changes in Sup Fig 7d and the C/EBPbeta enhancer hypomethylation.
Author Response	We agree with the reviewer that molecular changes such as C/EBPbeta enhancer hypomethylation may have started early at 4-old-month male TG mice at various degrees. We have analyzed the correlations of the molecular changes with the C/EBPbeta enhancer hypomethylation and expression levels. We did not observe significant correlations between C/EBPbeta enhancer hypomethylation and the expressions of the HCC driver genes (data not shown). However, the expressions of these genes were significantly correlated with C/EBPbeta expression (Supplementary Fig. 7e). One potential reason is that these C/EBPbeta target genes may not be solely regulated by C/EBPbeta enhancer methylation. C/EBPbeta over-expression by enhancer hypomethylation may cooperate with other early premalignant events to activate these oncogenes.
Excerpt from revised manuscript	(P.9-10:) Notably, these HCC driver genes were not only suppressed in C/EBPβ enh^{-/-} cells (p<0.05; Fig. 6i,j), but also up-regulated and correlated with C/ebpβ expression in the pre-malignant liver tissues of the HBx TG HCC model (p<0.05; Supplementary Fig. 7d,e). (P.11:) Given the concordant up-regulation of the oncogene orthologs (Fgfr2, Hif1a, Ralb, Rara) in HBx TG mouse model, which may be activated by C/ebpβ and other premalignant events, our results suggest a paradigm of enhancer activation of oncogenic cell signaling networks for further investigation. (Please see also the new Supplementary Fig. 7e.)

Ref 1.5. Effect of *C/EBPbeta* enhancer and *C/EBPbeta* itself on global enhancer activity

Reviewer	5. Fig. 6. The authors describe that C/EBPbeta enhancer affected global enhancer
----------	---

Comment	activity. This may be true, but it is not clear why they can conclude the changes observed were due to C/EBPbeta enhancer, not due to C/EBPbeta itself. Both C/EBPbeta enhancer deletion and eRNA/mRNA knock-down down-regulate C/EBPbeta, and the former is likely to have a stronger effect. If the authors want to keep the conclusion of C/EBPbeta enhancer, not due to C/EBPbeta itself, they need to compare the C/EBPbeta expression in Fig. 6a by adding eRNA/mRNA knock-down cells.
Author Response	We would like to clarify that in our original manuscript, we meant to say that C/EBPbeta enhancer may influence global enhancer activity through its regulation of C/EBPbeta. Besides, the incomplete eRNA/mRNA knockdown (see also our response to Ref 1.10 below) may compromise data validity or comparability with enhancer deletion. We have therefore modified the sentences involved to make it clear.
Excerpt from revised manuscript	(P.9:) Given the importance of C/EBPβ on global enhancer regulation⁴², we speculated that C/EBPβ enhancer may influence genome-wide enhancer activity through its regulation of C/EBPβ. (P.9:) Altogether, these findings indicate that C/EBPβ enhancer deletion impairs global enhancer activity via C/EBPβ/BRD4 dysregulation.

Ref 1.6. Use of DNA hypomethylating agents

Reviewer Comment	Minor Comments 1. This study rings an alarm on the use of DNA hypomethylating agents for cancers in which hypomethylation is a driving vicious positive feedback loop. The authors may want to discuss the issue.
Author Response	We have added some discussions of this important point in the revised manuscript.
Excerpt from revised manuscript	(P.12:) Like other cancers, HCC is characterized by global DNA hypomethylation, promoter hypermethylation, and more recently, hypomethylation associated with tumor-promoting gene up-regulation^{6, 68}. Together with our discovery of aberrant enhancer hypomethylation that drives vicious positive feedback circuitry, cautions should be taken for the use of DNA hypomethylating agents as cancer therapeutics. The recent advancement in CRISPR/Cas9-mediated epigenome editing^{31, 69} may offer more specific treatment strategies via enhancer reprogramming.

Ref 1.7. Clarification of relative gene expression

Reviewer Comment	2. Throughout the manuscript. The authors use "relative GENE expression" for quantitative RT-PCR (for example, Fig. 1j, and 2b-2h). The meaning of "relative" seems to be different depending upon panels. For 1j and 2b, it appears that delta-Ct values are displayed while the expression levels were normalized to a specific
------------------	---

	sample in Fig. 2c-2h. The meaning needs to be clarified.
Author Response	We have now clarified the meanings of relative gene expression in the corresponding figure legends (Fig. 1j-k, 2b, d-j, 3c-d, 4c, 5c-d and 6i-j) on P.21-23.

Ref 1.8. Merging of figures

Reviewer Comment	3. Fig. 7. The importance of this Fig. does not appear as high as that of the other Figs, and the authors may want to consider merging this one with Fig. 6 and moving some into a Supplementary Figs.
Author Response	We have merged Fig. 6 and 7 into one figure.
Excerpt from revised manuscript	(Please also see the updated Fig. 6.)

Ref 1.9. Revision of Supplementary Figure 4

Reviewer Comment	4. Supplementary Fig. 4. Vertical strips in red and green are misleading. The authors should simply mark the positions of the promoter and enhancer in the genome structure at the top.
Author Response	We have changed the illustrations of promoter and enhancer regions in Supplementary Fig. 4 from having vertical strips to the marks at the top of genome structure.
Excerpt from revised manuscript	(Please see the revised Supplementary Fig. 4.)

Ref 1.10. The degrees of eRNA knock-down

Reviewer Comment	5. Supplementary Fig. 6a. The degrees of eRNA knock-down need to be analyzed by qRT-PCR.
Author Response	The degrees of eRNA knockdown (~50-70% reduction as determined by qRT-PCR) have been shown in Fig. 2c of the original manuscript.

Ref 1.11. Description on the HBx-TG mice

Reviewer	6. Description on the HBx-TG mice is lacking in the "Methods" or "Supplementary
----------	---

Comment	Methods" section.
Author Response	We have added the description on the HBx TG mice in the Methods section.
Excerpt from revised manuscript	(P.15:) The experimental use of all mice was approved by the Animal Experimentation Ethics Committee of the Chinese University of Hong Kong. All mice received humane care according to the criteria outlined in the Guide for the Care and Use of Laboratory Animals (NIH). The strain of HBx TG mice was fixed to C57BL/6 by backcrossing with the C57BL/6 strain for more than 20 generations³⁷. Male HBx TG and WT mice were sacrificed at either 4 or 10 months of age, and the liver tissues were excised and snap-frozen.

Ref 1.12. Term correction

Reviewer Comment	7. Page 7, line mark 170. "5-aza-2-deoxycytidine (5'aza-dC)" -> "5-aza-2'-deoxycytidine (5-aza-dC)"
Author Response	We have amended the drug name accordingly.

Ref 1.12. Correction of typo

Reviewer Comment	8. Page 7, line mark 180, and other places. "5'-aa-dC" -> "5-aza-dC"
Author Response	We have amended the typo accordingly.

Ref 1.13. Phrase clarification

Reviewer Comment	9. Page 10, line marks 287-288. The authors may want to clarify the phrase "mechanisms underlying dysregulation remain unclear" by adding "from a DNA methylation viewpoint", and noting that the inheritance mechanism of DNA methylation is well established.
Author Response	We have clarified and incorporated the phrases accordingly.
Excerpt from revised manuscript	(P.10:) However, the mechanisms underlying enhancer dysregulation, especially from a DNA methylation viewpoint, remain unclear. (P.10:) Given the well-established inheritance mechanism of DNA methylation, this study provides evidence for the heritability and causality of enhancer alterations in cancer development.

Referee 2

Ref 2.1. C/EBP β enhancer specific hypomethylation

Reviewer Comment	This study can significantly contribute to uncover tumorigenesis of HCC. I sustain your enforcement. However I think your paper has some remediation points and propose following amendment points. Major #1. You demonstrated 5-aza-2dC treatment to make hypomethylation at C/EBPβ enhancer lesion and suggested that C/EBPβ enhancer hypomethylation is related to eRNA reactivation and expression. However, 5-aza-2dC induces genome wide hypomethylation, not specific at C/EBPβ enhancer. Therefore, in the results, we are unable to disregard influences of other sites hypomethylation. I will give you a suggestion that you should make C/EBPβ enhancer specific hypomethylation using epigenetic editing technology such as CRISPR/CAS9 system (DOI: 10.1002/anie.201601708, DOI: 10.1038/nbt.3658 etc.). If you make this specific hypomethylation model, you can make sure the result that C/EBPβ enhancer hypomethylation is related to HCC carcinogenesis. This model although help you to figure out the specific enhancer hypomethylation-associated C/EBPβ over-expression when you demonstrate exhausted analysis of expression variation of eRNA or mRNA, changes of methylation profiles between before C/EBPβ over-expression and after it.
Author Response	In order to provide a specific hypomethylation model, we employed targeted DNA demethylation by a modified dCas9-TET1 hydroxylase fusion construct (Morita et al. 2016) as recommended by the reviewer. We designed sgRNA to target specific CpG demethylation at the 5' end of the C/EBPβ enhancer (Supplementary Fig. 2a), where differential methylation between non-tumor and tumor tissues was more evident (region 1 in Fig. 1g,h). In a methylation rich cell line, SK-Hep1 (Fig. 2b), we demonstrated that targeted demethylation of two CpGs in the C/EBPβ enhancer (~25%), which was similar to the extent by 5-aza-dC as shown in Fig. 2c, resulted in significant up-regulation of C/EBPβ eRNA and mRNA (p<0.001; Supplementary Fig. 3a,b).
Excerpt from revised manuscript	(P.7:) To exclude potential influences by other hypomethylated sites upon 5-aza-dC treatment, we performed targeted DNA demethylation by a modified dCas9-TET1 hydroxylase fusion construct³¹ and demonstrated that targeted demethylation of C/EBPβ enhancer increased C/EBPβ eRNA and mRNA expressions (p<0.001; Supplementary Fig. 3a,b). (P.14:) Targeted DNA demethylation The pPlatTET-gRNA2 vector for dCas9-GCN4 and scFv-TET1CD-GFP fusion protein expression was obtained from Addgene (#82559)³¹. A single guide RNA (sgRNA) expression vector was modified from MLM3636 (Addgene #43860), in which 20-bp sgRNAs designed by CRISPR/Cas9 Target Online Predictor (CCTop)⁷⁰ were cloned. The

	sgRNA sequences for a control region⁶⁹ and C/EBPβ enhancer are 5'-CCCCCGGGGAAAAATTTT-3' and 5'-CACACACACAGGGCCACCGA-3', respectively. The pPlatTET-gRNA2 and sgRNA-expressing vectors were co-transfected into SK-Hep1 cells by jetPRIME from Polyplus Transfection according to the manufacturer's instructions. After 48 h, transfected cells were flow-sorted to isolate GFP-positive cells, followed by cell expansion for pyrosequencing and qRT-PCR analyses. (Please see also the new Supplementary Fig. 3a,b.)
--	---

Ref 2.2. Comparative investigation of eRNA KD in both methylation rich and poor cell lines

Reviewer Comment	#2. To indicate relationship between C/EBP β enhancer hypomethylation and eRNA and mRNA expression, you should do eRNA KD examination using both methylation rich cell lines and poor cell lines, and comparative investigation.
Author Response	We have performed additional siRNA-mediated knockdown of eRNA in the methylation rich cell lines, PLC5 and SK-Hep1, followed by MTT and invasion assays. We found that eRNA knockdown in these methylation rich cell lines reduced cell growth, but the extent of reduction (3-12%, Supplementary Fig. 6c) was less than that of the methylation poor cell lines, HepG2 and LO2, as shown in the original manuscript (24-28%, Supplementary Fig. 6a). For cell invasiveness, the results were consistent, in which the reduction in methylation rich cell lines (23-30%, Supplementary Fig. 6d) was clearly smaller than that in the methylation poor cell lines (40-53%, Supplementary Fig. 6b). This comparative investigation suggests that C/EBPβ enhancer methylation level may regulate cancer cell phenotypes via C/EBPβ eRNA. Cancer cells with C/EBPβ enhancer hypomethylation would be more dependent on the highly-expressed C/EBPβ eRNA/mRNA expression in comparison than those with enhancer hypermethylation.
Excerpt from revised manuscript	(P.8:) We found that down-regulation of C/EBPβ eRNA in HepG2 and LO2 cells significantly reduced cell growth and invasiveness ($p < 0.01$; Supplementary Fig. 6a,b), and the extents of reduction appeared to be more than PLC5 and SK-Hep1 cells whose C/EBPβ enhancer was hypermethylated (Supplementary Fig. 6c,d). As expected, stable C/EBPβ knockdown phenocopied the effects of eRNA knockdown (Supplementary Fig. 6e,f). Taken together, these data suggest that C/EBPβ enhancer methylation may regulate cancer cell phenotypes via C/EBPβ eRNA/mRNA expression. (Please see also the new Supplementary Fig 6c,d.)

Ref 2.3. Relationship between C/ebp β hypomethylation and tumorigenicity in HBx TG mice

Reviewer	#3. At HBx TG mouse examination, your results only suggest the relationship
----------	---

Comment	between HBx TG and trends of C/ebp β hypomethylation, evaluation of eRNA, mRNA and protein levels. These results are feeble to suggest that C/EBP β dysregulation by enhancer hypomethylation occurs early in the carcinogenic cascade and plays a crucial role for HCC tumorigenicity.
Author Response	We agree with the reviewer and have amended the statement.
Excerpt from revised manuscript	(P.9:) These data suggest that C/EBPβ dysregulation by enhancer hypomethylation is a molecular event preceding the onset of HCC development in HBx TG mice.

Ref 2.4. Reason for selection of C/EBP β enhancer

Reviewer Comment	Minor Introduction #1. At introduction, I hope you demonstrate the reason you focused on C/EBP β , or preliminary data previously.
Author Response	In addition to gene function, C/EBPbeta enhancer was selected based upon another important reason not mentioned in the original manuscript, namely the number of enhancers that target a particular gene. Having multiple enhancers increase our confidence that the target genes were affected by differential enhancer methylation. C/EBPbeta had the largest number of enhancers (3 of them) on the list of 27 DMEs. The only other gene with 3 enhancers on the list is LITAF . The C/EBPbeta enhancer we selected had the highest eRNA-mRNA correlation (0.863) based on FANTOM5 among all the enhancers on the list coming from a gene with multiple enhancers.
Excerpt from revised manuscript	(P.5:) Based on the number of enhancers that target a particular gene and its potential functional significance, we selected a FANTOM5 enhancer region that is ~90-kb downstream to its target gene C/EBPβ (Fig. 1f), a key hepatocyte transcription factor for liver regeneration^{22, 23}, for in-depth analysis. With multiple enhancers (3 out of 27) targeted C/EBPβ and a high eRNA-mRNA correlation of 0.863 across 808 FANTOM5 samples, the selected highly-confident enhancer-target pair exhibited significant hypomethylation-associated gene up-regulation in HCC tumors (Supplementary Table 1).

Ref 2.5. Off-target effect of eRNA knockdown experiment

Reviewer Comment	Result #2. At eRNA knock down experiment using siC/EBP β 1 and 2, do you state flatly that mRNA downregulation is not involved in off-target effect?
Author Response	We found that both siC/EBP β 1 and 2 against the C/EBPβ eRNA did not influence the expression of neighboring genes, implying no off-target effect. We have now

	included this notion in the Results.
Excerpt from revised manuscript	(P.7:) Intriguingly, knockdown of C/EBPβ eRNA in both lines reduced mRNA levels of C/EBPβ (Fig. 2e,f). We observed no change in the expression of the neighboring genes SMIM-25 and DPM1 located upstream and downstream of the C/EBPβ enhancer (Supplementary Fig. 2 and 3c), implying no off-target effect.

Ref 2.6. Clarification of the standards of cutoff values

Reviewer Comment	#3. At figure 1,m, you should clarify standards of cutoff values by which you classified HCC patients into strong group and weak group.
Author Response	The figure legend has been modified to clarify the cutoff criteria i.e. median.
Excerpt from revised manuscript	(P.21:) Kaplan-Meier survival analysis of 48 HCC patients according to their C/EBPβ hypomethylation status (relative methylation of tumor vs. non-tumor). Patients with strong hypomethylation (above median, n=24) show poorer (l) overall and (m) disease-free survival rates than those with weak hypomethylation (below median, n=24).

Ref 2.7. Clarification of the information of HCC patients

Reviewer Comment	Methods #4. You should clarify information of patient's characters, especially background of liver diseases. Readers are concerned about result when you classify patients according to causes of HCC such as HBV infection, HCV infection and NAFLD.
Author Response	The information of the HCC patients, including the status of HBV/HCV infection and NAFLD, has now been provided.
Excerpt from revised manuscript	(P.13:) HCC patients who underwent hepatectomy at the Prince of Wales Hospital (Hong Kong) were included in this study (Supplementary Table 2). (Please see also the new Supplementary Table 2.)

Ref 2.8. Correction of typo

Reviewer Comment	figures #5. At figure3c, you should correct siRNA name such as following. siC/EBBβ-1→siC/EBPβ1
Author Response	The typo has been corrected.

References

- Heyn, H., E. Vidal, H. J. Ferreira, M. Vizoso, S. Sayols, A. Gomez, S. Moran, R. Boque-Sastre, S. Guil, A. Martinez-Cardus, C. Y. Lin, R. Royo, J. V. Sanchez-Mut, R. Martinez, M. Gut, D. Torrents, M. Orozco, I. Gut, R. A. Young, and M. Esteller. 2016. "Epigenomic analysis detects aberrant super-enhancer DNA methylation in human cancer." *Genome Biol* 17:11. doi: 10.1186/s13059-016-0879-2.
- Mazor, T., A. Pankov, J. S. Song, and J. F. Costello. 2016. "Intratatumoral Heterogeneity of the Epigenome." *Cancer Cell* 29 (4):440-451. doi: 10.1016/j.ccell.2016.03.009.
- Morita, S., H. Noguchi, T. Horii, K. Nakabayashi, M. Kimura, K. Okamura, A. Sakai, H. Nakashima, K. Hata, K. Nakashima, and I. Hatada. 2016. "Targeted DNA demethylation in vivo using dCas9-peptide repeat and scFv-TET1 catalytic domain fusions." *Nat Biotechnol* 34 (10):1060-1065. doi: 10.1038/nbt.3658.

REVIEWERS' COMMENTS:

Reviewer #1 (Remarks to the Author):

Although it was a pity that the authors did not modify the manuscript for comment #3, all the other issues have been adequately modified. This reviewer does not have any additional comments.

Reviewer #2 (Remarks to the Author):

In this revised manuscript, the authors responded well to our comments.

Editor

Editorial Comment	Given the concerns of the reviewers regarding the strength of evidence supporting carcinogenesis please amend the Abstract, Title, and Discussion to accordingly.
Author Response	We thank the editor for the comment. We have tuned down our statements on the relationship between C/EBP β hypomethylation and carcinogenesis.
Excerpt from revised manuscript	Title: (P.1:) Aberrant Enhancer Hypomethylation Contributes to Hepatic Carcinogenesis through Global Transcriptional Reprogramming Abstract: (P.2:) deletion of this enhancer via CRISPR/Cas9 reduces C/EBPβ expression and its genome-wide co-occupancy with BRD4 at H3K27ac-marked enhancers and super-enhancers, leading to drastic suppression of driver oncogenes and HCC tumorigenicity... These results support a causal link between aberrant enhancer hypomethylation and C/EBPβ over-expression, thereby contributing to hepatocarcinogenesis through global transcriptional reprogramming. Discussion: (P.11:) Given the concordant up-regulation of the oncogene orthologs (Fgfr2, Hif1a, Ralb, Rara) in HBx TG mouse model, which is correlated with C/EBPβ, our results suggest a paradigm of enhancer regulation of oncogenic cell signaling for further investigation.